# FlhE functions as a chaperone to prevent formation of periplasmic flagella in Gram-negative bacteria

Manuel Halte [1] ✉, Ekaterina P. Andrianova [2], Christian Goosmann[3], Fabienne F. V. Chevance [4], Kelly T. Hughes [4], Igor B. Zhulin [2] & Marc Erhardt [1,5] ✉

The bacterial flagellum, which facilitates motility, is composed of ~20 structural proteins organized into a long extracellular filament connected to a cytoplasmic rotor-stator complex via a periplasmic rod. Flagellum assembly is regulated by multiple checkpoints that ensure an ordered gene expression pattern coupled to the assembly of the various building blocks. Here, we use epifluorescence, super-resolution, and transmission electron microscopy to show that the absence of a periplasmic protein (FlhE) prevents proper flagellar morphogenesis and results in the formation of periplasmic flagella in *Salmonella enterica*. The periplasmic flagella disrupt cell wall synthesis, leading to a loss of normal cell morphology resulting in cell lysis. We propose that FlhE functions as a periplasmic chaperone to control assembly of the periplasmic rod, thus preventing formation of periplasmic flagella.

The bacterial flagellum is a complex nanomachine whose assembly and regulation are finely tuned in order to achieve functional assembly. As a rotatory structure, the flagellar filament enables bacteria, including the Gram-negative bacterium *Salmonella enterica* serovar Typhimurium (*S. enterica*), to swim into their environment for nutrient acquisition or evasion of harmful substances[1,2]. The flagellum structure includes three main parts: a basal body that includes a drive shaft (rod) that spans the periplasmic space and several protein rings embedded into the inner membrane (IM), peptidoglycan layer (PG), and outer membrane (OM); a flexible hook outside the cell; and a rigid filament made up of as many as 20,000 flagellin units[3]. Flagellum assembly is a tightly controlled process governed by specific genetic regulatory mechanisms and hierarchical order of component assembly[4,5]. Expression of the flagellar master regulatory complex FlhD$_4$C$_2$ is integrated to environmental signals and is essential to activate the expression of genes encoding proteins required for the structure and assembly of the flagellum structure[6] (Fig. 1a). Basal body assembly initiates in the IM, forming the export gate (FliPQR/FlhBA) followed by

the MS-ring (FliF) and C-ring (FliGMN) involved in the rotation[7,8]. After IM rings assembly, the rod components are secreted through the export gate of the flagellar type-III secretion system (fT3SS) and begin to assemble in the periplasm, forming the proximal rod (FliE/FlgBCF) and distal rod (FlgG). Polymerization of the rod relies on FlgJ, a dual-function protein. FlgJ serves as a cap to facilitate rod formation and digests the PG layer, allowing the distal rod to polymerize to the OM. Two additional rings of protein, the P-ring (FlgI) and L-ring (FlgH), embedded into the PG and OM layer respectively, are required to assemble a functional extracellular flagellar filament[3,9]. Upon assembly of the L-ring, the structure continues to grow into the extracellular environment, as the hook subunits (FlgE) polymerize onto the tip of the distal rod. Once the hook reaches a length of ~55 nm, the molecular ruler FliK determines a minimal length of the entire rod-hook structure and catalyzes a switch of substrate specificity within the export gate from early (or hook-basal body (HBB) subunits) to filament subunits (e.g. flagellins), which after secretion, polymerize into the long helical filament[10,11].

[1]Institute of Biology, Humboldt-Universität zu Berlin, Philippstr. 13, 10115 Berlin, Germany. [2]Department of Microbiology, The Ohio State University, Columbus, OH 43210, USA. [3]Max Planck Institute for Infection Biology, Charitéplatz 1, 10117 Berlin, Germany. [4]School of Biological Sciences, University of Utah, Salt Lake City, UT, USA. [5]Max Planck Unit for the Science of Pathogens, Charitéplatz 1, 10117 Berlin, Germany. ✉e-mail: manuel.halte@hu-berlin.de; marc.erhardt@hu-berlin.de

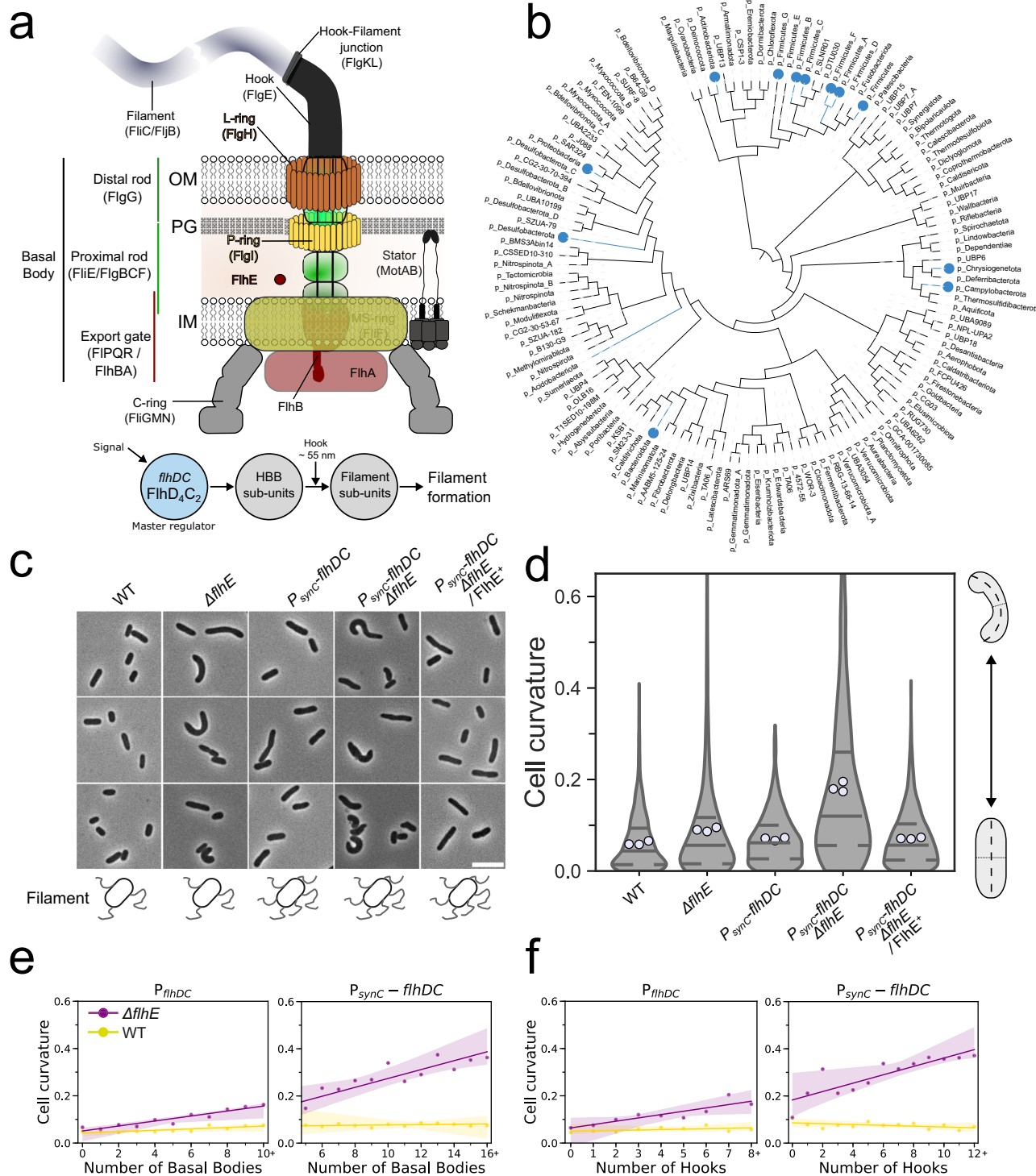

**Fig. 1 | FlhE: genetic distribution and effect on cell shape curvature. a** Schematic representation of the fT3SS structure and order of assembly after expression of the master regulator *flhDC*. **b** Genome tree of bacteria shown at the phylum level, adopted from the GTDB database. Phyla where the FlhE domain was identified are marked with a blue dot. **c** Exemplary images of stationary phase WT and *ΔflhE* mutant cells observed in phase contrast microscopy. *ΔflhE* mutant displays an aberrant morphology. Overexpression of *flhDC* using a strong constitutive promoter ($P_{synC}$) increased the shape defect and complementation of *flhE* in trans restored the WT phenotype. Scale bar = 5 μm. Schematics below the microscopy images display the flagellation pattern of the WT and $P_{synC}$. **d** Cell curvature analysis of the cells displayed in **c**. Measurements of the cell curvature were performed using MicrobeJ[59]. 3 independent biological replicates were performed. Each white

dot represents the mean curvature of one biological replicate. The violin plot represents the median and inner quartiles of the whole population. At least 300 cells per replicate were analyzed. **e** Cell curvature increase in the *ΔflhE* mutant is correlated with an increase in number of basal bodies (BB, FliG-mNeonGreen) and **f** hooks (FlgE$_{S171C}$, Maleimide STAR RED) in a WT *flhDC* background. Similar observations were made with a constitutively expressed $P_{synC}$-*flhDC* background. 3 independent biological replicates were performed. At least 250 cells per replicate were analyzed using MicrobeJ. Dots represent the cell curvature mean for all cells with a given number of basal bodies or hooks. The curve represents the linear regression line with an interval confidence CI = 99.9 (WT in yellow, *ΔflhE* mutant in magenta). Source data are provided as a Source Data file.

Of the forty flagellar gene products known to be involved in the structure and assembly of the bacterial flagellum and coupled gene regulatory mechanisms, FlhE is one of the few remaining proteins whose function in flagellum assembly has remained elusive. FlhE is a periplasmic Sec-translocated protein expressed from the *flhBAE* operon. The other two gene products FlhB and FlhA are integral components of the fT3SS export gate. The role of FlhE in regulation and assembly of the fT3SS and the flagellum remained unclear. Previous studies have demonstrated that deletion of *flhE* does not affect swimming motility or the number of flagella[12–14]. Interestingly, however, the absence of FlhE resulted in infrequent cell lysis that was dependent on assembly of flagellar filaments. For colonies grown on green indicator medium (green plates, GP), cell lysis results in a blue color phenotype, which is normally used as a visual assay for phage-infected cells[12]. However, the relation between FlhE loss, flagellar filament assembly, and cell death remained unclear. Consequently, we sought to characterize the mechanism, dynamics, and evolution of FlhE to clarify the role of this seemingly "non-essential" flagellar gene in flagellar filament assembly.

In this study, we observed that FlhE plays a role in maintaining the rod cell shape of *S. enterica* under conditions when flagellar filaments are assembled. Without FlhE, key cytoskeleton structures required for proper PG synthesis were misplaced, causing the cells to lose their rod cell morphology. We found that this mis-localization of the divisome and elongasome complexes was caused by the ectopic assembly of flagellar filaments in the periplasm in the absence of FlhE. Our results suggest that FlhE functions as a periplasmic chaperone to control the assembly of the flagellar rod until formation of the PL-rings and penetration of the OM, thereby preventing formation of periplasmic flagella, which would otherwise disrupt PG synthesis resulting in abnormal cell morphology and ultimately cell lysis.

## Results

### FlhE was recently recruited to the flagellar system of Gammaproteobacteria

FlhE is present in Gammaproteobacteria *S. enterica* and *E. coli*[12,15], but it was not identified as a part of the flagellar regulon in other model organisms representing diverse bacteria phyla, such as *Bacillus subtilis* (Bacillota)[16], *Helicobacter pylori* and *Campylobacter jejuni* (Campylobacterota)[17], *Borrelia burgdorferi* and other spirochetes[18]. Furthermore, FlhE was not reported as a part of flagellar systems in closely related Gammaproteobacteria, such as species of *Vibrio*[19] and *Pseudomonas*[20]. To reveal how this protein became a part of the flagellar regulon in the peritrichously flagellated enteric bacteria *S. enterica* and *E. coli*, we first searched for FlhE homologs in a dataset of representative bacterial genomes from the Genome Taxonomy Database[21] (for details, see Materials and Methods). As a result, we identified FlhE domain-containing proteins in representatives of a dozen bacterial phyla with sporadic distribution across the bacterial tree of life (Fig. 1b and Supplementary Dataset 1). We then analyzed gene neighbourhoods of *flhE* homologs and found that *flhE* genes are located in flagellar operons only in representatives of three Gammaproteobacterial orders: *Enterobacteriales*, *Pseudomonadales*, and *Burkholderiales* (Fig. S1a and Supplementary Dataset 1). Remarkably, the gene order in *flhE*-containing operons is conserved: *flhE* always follows *flhB* and *flhA* (in some cases interrupted with *flhF* and *flhG*) and the flagellar operon is preceded by the chemotaxis operon of the evolutionary class F7[22]. Because of this conservation, in all organisms where *flhE* is a part of the conserved flagellar/chemotaxis operon, FlhE is predicted to play the same role in other bacteria as it does in *S. enterica* and *E. coli*. However, the role of FlhE in flagellar systems of other organisms is unclear. Intriguingly, the flagellar filament outer layer protein FlaA, which is specific to the periplasmic flagella of spirochetes, is also a member of this superfamily. Although we cannot establish homology at the sequence level between FlaA and FlhE, their structural similarity is remarkable (Fig. S1b).

### Assembly of flagellar filaments results in cell lysis in the absence of FlhE

Consistent with the finding by Lee et al.[12], *S. enterica* mutants lacking *flhE* exhibited a cell lysis phenotype on GP (blue colonies) under conditions when flagellar filaments were assembled (Fig. S2a). Complementation of a *flhE* deletion mutant with *flhE* expressed in trans restored the wildtype (WT) phenotype (white colonies). In addition, a mutation in any component of the fT3SS (ΔfliPQR/ΔflhBA), the rod (ΔflgBC), the hook (ΔflgE) or the hook-filament junction (ΔflgKL), which prevented assembly of flagellar filaments, restored the WT phenotype in the ΔflhE background (Fig. S2b). Reducing the length of the assembled flagellar filament by deleting the flagellin secretion chaperone FliS[23] restored the WT phenotype on GP. fT3SS mutants locked in early-mode secretion, by either using the presence of an allele of FlhB unable to undergo auto-catalytic cleavage (FlhB$_{N269A}$)[24], or a deletion of the gene of the molecular ruler FliK, also presented a WT phenotype on GP. Finally, blocking secretion of substrates via the fT3SS using a non-translocatable FliC-GFP translational fusion[25,26] resulted in a WT phenotype on GP. Cell lysis was independent of the flagellin composition of the filament (FliC-ON/FljB-ON)[27,28], as well as filament rotation as a deletion (ΔmotAB) or non-functional mutant (MotA$_{M206A}$/MotB$_{D33A}$) of the stator units did not restore the WT phenotype[29] (Fig. S2b). Lastly, a deletion of the response regulator CheY, which results in a straight-swimming phenotype, did not restore the WT phenotype in the ΔflhE background, demonstrating that the rotational direction of the flagellar filament was not responsible for the observed phenotype (Fig. S2b).

### Overproduction of flagellar filaments in the absence of FlhE results in an aberrant, curved cell morphology

Based on our findings that, in the absence of FlhE, preventing either assembly or reducing the length of the flagellar filament restored the WT phenotype on GP, we hypothesized that the cell lysis phenotype would be exacerbated by increasing the number of flagella per cell. We therefore investigated swimming motility, cell morphology and the lysis phenotype of ΔflhE strains expressing flagella at WT or elevated levels by replacing the native *flhDC* promoter with synthetic constitutive promoters of different strength (P$_{pro}$ series)[30], here named P$_{syn}$. Swimming motility assays revealed that WT and ΔflhE strains expressing *flhDC* from weak constitutive promoters (P$_{synA}$/P$_{synI}$) had reduced motility compared to the native *flhDC* promoter. Expression of *flhDC* from strong, constitutive promoters (P$_{synB}$ or P$_{synC}$) increased motility by approximately 1.5-fold in the WT background, yet decreased it in the ΔflhE background (Fig. S2c). At native levels of *flhDC* expression, cell lysis was not observed using phase contrast microscopy in the absence of FlhE when grown to exponential phase. However, a subset of cells (~3% of the population) from a 16-hour stationary phase culture showed a loss of phase contrast, indicative of a loss of the cytoplasm. Remarkably, some cells presented an aberrant cell morphology distinct from the rod cell shape of the WT (Fig. 1c). Live/dead staining was then carried out comparing WT and ΔflhE under native *flhDC* and varied constitutive *flhDC* expression (decreased expression in P$_{synA}$, and overproduction in P$_{synC}$) (Fig. S2d). Slight increase in cell death ratios were observed with the native *flhDC* promoter for ΔflhE when compared to WT. However, cell death in the P$_{synC}$-*flhDC* ΔflhE background increased, with roughly 30% of cells stained with propidium iodide. No such increase was observed with the P$_{synA}$ promoter. We concluded that the increase in cell death accounts for the observed decrease in motility in the P$_{synC}$-*flhDC* ΔflhE background.

Measurement of the cell curvature revealed a loss of the standard rod-shape in the ΔflhE mutant, and the proportion of cells with an

aberrant morphology was exacerbated in the $P_{synC}$ background (Fig. 1c, d). The $P_{synC}$-flhDC ΔflhE strain further exhibited a blue phenotype on GP (Fig. S2b), which was restored by ectopic FlhE$^+$ expression. The $P_{synC}$-flhDC ΔflhE strain also exhibited slower growth in liquid culture and smaller colonies compared to the $P_{synC}$-flhDC strain on standard LB agar plates. Consistent with observations by Lee et al.[12], a deletion of fliS, resulting in shorter filament, restored the WT phenotype; however, overexpression of FlhDC in the ΔflhE ΔfliS background caused the blue phenotype on GP (Fig. S2b). We also observed that deletion of the structural genes coding for the hook-filament junction proteins FlgKL restored the WT rod-shape morphology (Fig. S2e, f). To confirm that the ΔflgKL deletion had no polar effect on the assembly of the flagellar filament, we expressed the WT flgKL genes under the chromosomal, arabinose-inducible, araBAD promoter (FlgKL$^+$). In absence of arabinose cells lack assembled filaments and no defect in cell shape was observed after overnight culture incubation. In contrast, the culture supplemented with arabinose displayed cells with aberrant morphologies. Using the same araBAD expressing system, we expressed a FliC-GFP fusion, which is not secreted via the fT3SS and does not assemble filament. A mutant of flhE in this background did not display a changed morphology or growth in the presence of arabinose, suggesting that such a system could be used to control the timing of the cell morphology defect caused by the absence of FlhE (Fig. S2e, f).

We next determined the number of basal bodies (BB, FliG-mNeonGreen) and hooks (FlgE$_{S171C}$) in the WT and ΔflhE background under various flhDC expression conditions. We observed a correlation between the increase in cell curvature and the numbers of BB and hook structures in the absence of FlhE, but not in the presence of FlhE. Those observations were irrespective of the FlhDC expression level, using either the native flhDC or synthetic constitutive $P_{synC}$ promoter (Fig. 1e, f), or in the presence of an anhydrotetracycline (AnTc) inducible $P_{tetA}$ promoter ($P_{tetA}$-flhDC)(Fig. S3).

## Aberrant cell morphology in the absence of FlhE is caused by divisome and elongasome mis-localization

We hypothesized that the defects in cell morphology observed in the ΔflhE mutant were associated with irregular cell wall formation as maintenance of cell shape during division or elongation relies on proper PG assembly and turnover[31,32]. In this respect, mis-localization of essential components of the cell division process, including FtsZ, MinD, and MreB were previously observed in cells where the proton motive force (PMF) was disrupted in both *E. coli* and *Bacillus subtilis*[33]. We note that the absence of FlhE has previously been associated with acidification of the cytoplasm, possibly linked with proton leakage[12]. Hence, we speculated that the observed aberrant cell morphology phenotype might be indirectly caused by Z-ring mis-localization, resulting from a general defect in preserving the membrane potential in the ΔflhE background. To test this hypothesis, we examined Z-ring formation in $P_{synC}$-flhDC strains in the presence or absence of FlhE using a non-functional FtsZ-eGFP (C-terminal) expressed in trans from a leaky IPTG-inducible plasmid[34]. Time-lapse microscopy on an agarose pad revealed Z-ring mis-localization from middle cell to an ectopic position in a $P_{synC}$-flhDC ΔflhE mutant, unlike in $P_{synC}$-flhDC FlhE$^+$ cells where all Z-rings localized to the middle cell (Fig. 2a). Further, we note that release of cytoplasm content observed in phase contrast appeared to coincide with the ectopic Z-ring position (Fig. S4a). Aberrant Z-ring localization in the ΔflhE mutant was also visible using confocal microscopy, with multiple incompletely formed Z-rings stretching across the whole cell (Figs. 2b and S4b). We ruled out chromosome segregation defects by combining Z-ring visualization and DNA staining (Fig. S5a). Chromosome localization correlated with the Z-ring localization in the cells exhibiting aberrant cell morphology, hinting at mis-regulated PG synthesis.

We next performed killing curve assays in the presence of antibiotics that impact either cell wall formation (ampicillin/cephalexin) or mRNA translation (kanamycin). Cephalexin and ampicillin (Ap) are β-lactam antibiotics that bind and inactivate penicillin-binding proteins (PBPs) involved in PG synthesis during cell growth and division[31]. We hypothesized that inhibiting PG synthesis, and thereby putting an additional strain on the PG layer, would exacerbate the cell death phenotype of a ΔflhE mutant. Killing curves of strains in the $P_{synC}$-flhDC ΔflhE background showed a higher sensitivity towards cell wall targeting antibiotics compared to $P_{synC}$-flhDC parental strain (Fig. S5b). For the translation-inhibiting antibiotic kanamycin, no differences in sensitivities were detectable. Along these lines, direct observation of cells after treatment using phase contrast microscopy revealed a decreased contrast. This is characteristic of dead cells after ampicillin and cephalexin treatment, suggesting a higher sensitivity to PG-targeting drugs in the ΔflhE background over WT (Fig. S5c).

Bacteria can survive a cell wall defect for several hours in the absence of permeabilization of the OM due to the mechanical stabilization role of the OM[35]. This could explain why most of ΔflhE cells from a stationary phase culture were still able to divide despite having a cell shape defect. We stained the cells with SYTOX Green, a green nucleic acid binding dye only able to stain DNA in cells with pores in both the OM and IM[36,37]. In the ΔflhE mutant, cells with aberrant morphology did not show a SYTOX Green signal, suggesting that those cells with shape defect retained either an intact OM and/or IM (Fig. S5d). We next observed the growth of the ΔflhE mutant over time in a microfluidic device continuously supplied with fresh medium. As shown in Figs. 2c and S5e, the ΔflhE mutant continued to perform cell division, albeit with frequent cell morphology defects and minicell formation. Minicell formation is a hallmark of incorrect Z-ring localization and suggests a defect of the Min system, thereby failing to prevent cell division close to the cell poles[38].

Further evidence of PMF disruption in a flhE mutant was obtained by visualizing a MreB-mNeonGreen sandwich fusion between residues G228-D229. In a WT background, MreB assembled in proto-filaments encircling the cell, as observed before[39,40]. Yet, in a $P_{synC}$-flhDC ΔflhE background, MreB filament localization was disrupted, with visible fluorescence throughout the entire cell. Additionally, fluorescence intensity was around 10-folds weaker than in WT, suggesting a disassembly of elongasome complexes (Figs. 2d and S4c). Taken together, these observations demonstrate that loss of FlhE causes a defect in the spatial control of PG synthesis leading to an aberrant cell shape and eventually sets the path for cell lysis.

## Absence of PL-rings assembly in the ΔflhE background exacerbates cell death

While a deletion of the export gate (FliPQR), fT3SS (FlhBA) and proximal rod components (FlgBC) restored WT phenotypes in the absence of FlhE, we noted that deletion of structural genes for either the periplasmic ring (P-ring, flgI) or the outer-membrane ring (L-ring, flgH) was lethal. To circumvent the growth defect, we constructed a ΔflhE ΔflgHI double mutant in a $P_{tetA}$-flhDC background, where the flagellar regulon is expressed only in presence of tetracycline (Tc) or AnTc. On LB plates in the absence of inducer, the ΔflhE ΔflgH/I double mutant was fully viable (Fig. 3a). Upon addition of Tc, the viability of flhE mutants lacking PL-rings components (ΔflgH ΔflhE/ΔflgI ΔflhE/ΔflgHI ΔflhE) decreased drastically, while growth of $P_{tetA}$-flhDC ΔflhE and $P_{tetA}$-flhDC ΔflhE Δrod (ΔflgBC) was not affected in the presence of inducer. Similar to the observations made in the $P_{synC}$-flhDC ΔflhE background, assembly and localization of the divisome complex components ZapA (mNeonGreen-ZapA) showed ectopic localization in ΔflgHI ΔflhE background upon flhDC expression (Fig. S6a). mNeonGreen-MinD oscillations in ΔflgHI ΔflhE were also reduced overtime (Fig. S6b).

We were surprised that the absence of FlhE in a PL-rings mutant background was non-viable, since we previously established that cell

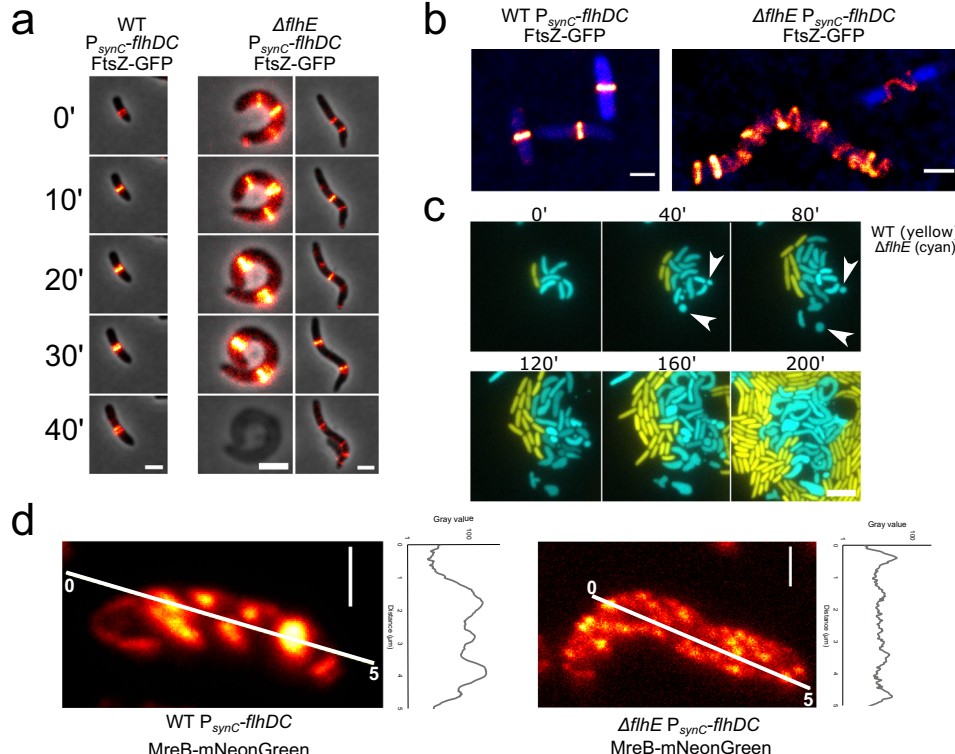

**Fig. 2 | The loss of FlhE impairs localization and assembly of the divisome and elongasome. a** Time lapse microscopy of plasmid-based expressed FtsZ-GFP in the $P_{synC}$-flhDC and $P_{synC}$-flhDC ΔflhE background. Two distinct cells with ectopic localisation of FtsZ are displayed for $P_{synC}$-flhDC ΔflhE background. Scale bar = 2 μm. **b** Maximum projection of a z-stack confocal microscopy image of FtsZ-eGFP expressed in the $P_{synC}$-flhDC and $P_{synC}$-flhDC ΔflhE background with 70 nm z-step interval. DNA staining with Maleimide 560 Live (Abberior) is represented in blue. Scale bar = 1 μm. **c** Formation of minicells in ΔflhE background. Microfluidic experiment (CellAsic ONIX, Merck) of $P_{synC}$-flhDC (yellow, mNeonGreen) and $P_{synC}$-flhDC ΔflhE (cyan, mCerulean) expressing fluorescent proteins from a constitutive

promoter carried on a vector. Cells were continuously injected with LB at 1 psi. $P_{synC}$-flhDC ΔflhE displays an exacerbated disruption of morphology compared to WT and observation made from a culture grown in a flask and observed on an agarose pad. White arrows show the formation of minicells in $P_{synC}$-flhDC ΔflhE, suggesting a defect in MinD and/or FtsZ function (scale bar = 5 μm). **d** Confocal microscopy of MreB-mNeonGreen in the $P_{synC}$-flhDC and $P_{synC}$-flhDC ΔflhE background with 70 nm step range z-stack after maximum intensity projection. Scale bar = 1 μm. Data shown are from one representative experiment and have been independently replicated in at least one preliminary experiment.

lysis and cell morphology defects of *flhE* mutants were dependent on flagellar filament assembly (Fig. S2). In addition, HBB completion, which is the prerequisite for the substrate specificity switch and thus class III gene expression and filament assembly, should not be possible in a PL-rings mutant background. To investigate if the absence of *flhE* allowed for class III gene expression in a PL-rings mutant, we used a chromosomal *fliC-mneongreen* transcriptional fusion to monitor class III gene expression in a time-course experiment using epifluorescence microscopy. WT and Δ*flhE* strains in the presence of the PL-rings (FlgHI⁺) demonstrated similar expression dynamics of the class III reporter, while as expected, a Δ*flgHI* strain showed no expression of class III genes. However, the Δ*flgHI* Δ*flhE* double mutant also expressed class III genes, albeit with a delay compared to the WT (Fig. 3b). We also observed aberrant cell morphology and increase in cell curvature in the Δ*flgHI* Δ*flhE* double mutant, similar to what we observed using the $P_{synC}$-*flhDC* background (Fig. 3c). Given that the FliC flagellin is expressed in the Δ*flgHI* Δ*flhE* double mutant, we postulated that the loss of FlhE could potentially trigger an early substrate specificity switch, leading to ectopic secretion of flagellin into the periplasm and a toxic effect on the cells, which might cause the observed cell morphology defects. We therefore performed immunostaining using anti-flagellin primary antibodies in order to visualize potential periplasmic flagellins. The immunostaining revealed assembled extracellular flagellar filaments in the Δ*flgHI* Δ*flhE* double mutant, suggesting that these filaments could either assemble outside the cell envelope or in the periplasm before traversing the OM, even in absence of the PL-

rings (Fig. 3d). If those filaments assemble in the periplasm, it is possible that the staining procedure used here did not permit their observation, notably due to the high molecular weight of antibodies. Extracellular flagellar filament assembly was also observed in either the P-ring (Δ*flgI*) or L-ring (Δ*flgH*) Δ*flhE* mutant background (Fig. S7a). To establish that neither *flgI* nor *flgH* deletions exerted a polar effect on the remainder of the *flg* operon (*flgBCDEFGHIJ*), we complemented these deletions using chromosomal $P_{araBAD}$ expression constructs (Fig. S7b) and found that the single deletions could be complemented to motility levels ranging from 60% to 85% relative to the WT.

### Filament formation in the absence of FlhE requires hook and rod assembly in a PL-rings mutant

Surprised by the formation of flagellar filaments in the Δ*flgHI* Δ*flhE* double mutant, we speculated that filament assembly would still necessitate the presence of the hook (FlgE), rod cap and PG-hydrolase (FlgJ), and the distal rod (FlgG). Indeed, deletions of these genes restored the WT growth phenotype on plates as observed by spot-assay (Fig. S7c), while complementation in trans (FlgE⁺/FlgG⁺/FlgJ⁺) reintroduced the reduced growth rates in the Δ*flgHI* Δ*flhE* double mutant background. We note that these deletions displayed a minor polar effect on the *flg* operon, as complementation did not fully replicate the cell death phenotype seen in the Δ*flgHI* Δ*flhE* double mutant. Furthermore, the substrate specificity switch and class 3 gene expression observed in the Δ*flgHI* Δ*flhE* mutant required the presence of the hook, as deletion of *flgE* prevented the expression of *fliC*.

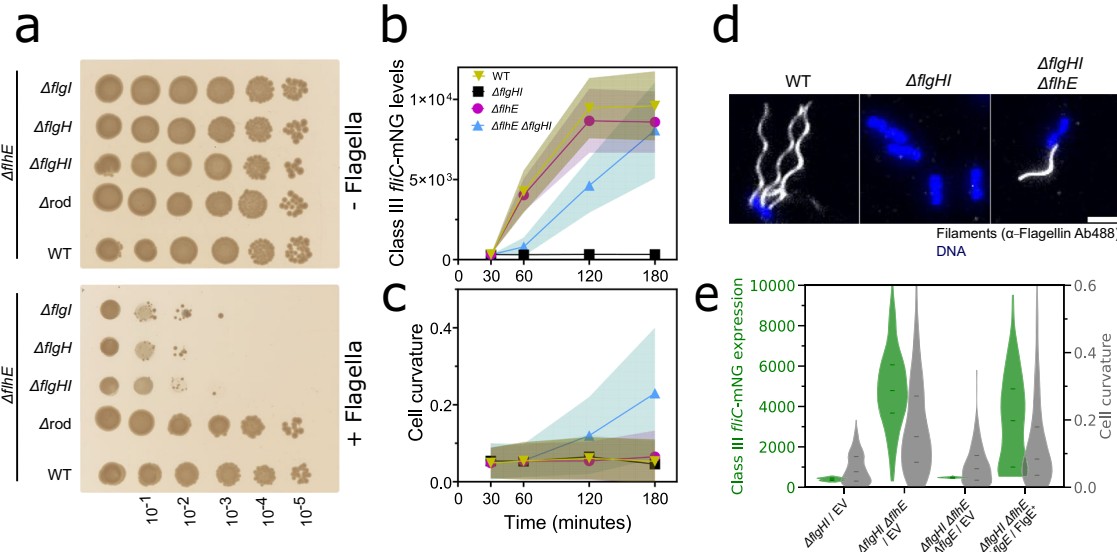

**Fig. 3 | Cell death is exacerbated in Δ*flhE* Δ*flgHI* background and requires flagellar filament assembly. a** Spot-assay of Δ*flhE* strains coupled with a deletion Δrod (Δ*flgBC*), P-ring (Δ*flgI*), L-ring (Δ*flgH*) or P-and L-rings (Δ*flgHI*). Δ*flhE* and Δ*flhE* Δrod showed no growth defect, while Δ*flgI*, Δ*flgH* and Δ*flgHI* could not grow following a 10⁻² dilution. The flagellar system (P_{tetA}:*flhDC*) is only expressed in presence of tetracycline inducer (+Flagella, Tc plate). In absence of inducer (−Flagella, LB plate), no growth defect can be observed for any of the tested strains. Data shown are from one representative experiment and have been independently replicated in at least three independent experiment. **b** Time course epifluorescence microscopy was carried out to probe the class III gene expression in strains carrying the P_{tetA}:*flhDC* and a class III reporter *fliC-mneongreen* transcriptional fusion, induced with AnTc and sampled after 30, 60, 120 and 180 min. The Δ*flhE* Δ*flgHI* mutant expresses class III gene, unlike the Δ*flgHI* strain. At least 185 cells were measured per strain and timepoint. **c** Measurement of the cell curvature of the

experiment performed in **b**. Shaded area represents the standard deviation. **d** Immunostaining of flagellar filament reveals that the Δ*flhE* Δ*flgHI* mutant can assemble filaments. Immunostaining was performed with primary anti-FliC antibodies and secondary antibodies coupled to Alexa488 (white). DNA staining was performed with Fluoroshield with DAPI (blue). Scale bar = 2 μm. **e** Expression of class III reporter *fliC-mneongreen* transcriptional fusion (green violin plot) in Δ*flgHI*, Δ*flhE* Δ*flgHI*, Δ*flhE* Δ*flgHI* Δ*flgE* and Δ*flhE* Δ*flgHI* Δ*flgE* complemented with pTrc-FlgE⁺, 180 min after P_{tetA}:*flhDC* induction with AnTc. No class III expression can be observed in Δ*flhE* Δ*flgHI* in absence of the hook (Δ*flgE*). Complementation in trans of *flgE* lead again to expression, confirming that the deletion has no polar effect on the *flg* operon. Grey violin plot represents the cell curvature, restored to WT level upon *flgE* deletion. In trans complementation caused again the curvature to increase. At least 300 cells per strain were analyzed. Source data are provided as a Source Data file.

Complementation of a Δ*flgHI* Δ*flhE* Δ*flgE* strain with WT *flgE* expressed in trans (FlgE⁺) restored the specificity switch and displayed an increase in cell curvature (Fig. 3e).

The absence of the flagellin chaperone FliS also reduced the growth defect observed on plate (Fig. S7c). Since a Δ*fliS* deletion mutant exhibits shorter filaments than the WT−a consequence of reduced flagellin transport to the export gate and therefore decreased flagellin secretion[23,41–43]−this result suggested that filament length was the primary determinant of cell death and abnormal cell morphology observed in the Δ*flgHI* Δ*flhE* double mutant. To confirm that the filament assembly rather than the secretion of flagellin alone into the periplasm was the cause for the observed toxicity, we expressed either a WT full-length FliC or a FliC truncated mutant deleted for its C-terminal domain required for flagellin polymerization[44] (Δaa451-495) in trans in a Δ*flgHI* Δ*flhE* Δ*fliC* strain. In absence of inducer and flagella formation, the Δ*fliC* strain displayed no growth defect (Fig. S7d), whereas complementation with full-length flagellin restored the growth defect in presence of flagella assembly (Δ*fliC*/*fliC*⁺). Complementation with the truncated FliC did not affect cell viability (Δ*fliC*/*fliC*⁺Δaa451-495), thereby confirming that filament assembly, and not only flagellin secretion, was necessary for the toxicity phenotype observed in the Δ*flgHI* Δ*flhE* mutant. Finally, we assessed the requirement of other components of the fT3SS for the substrate specificity switch observed in the Δ*flhE* background. We found that deletions of other rod components (Δ*flgBC*, Δ*flgI*, Δ*flgG*) prevented the substrate specificity switch in a Δ*flhE* strain (Fig. S7e).

Since the substrate specificity switch depends on secretion of the molecular ruler FliK[11,45], we hypothesized that FlhE facilitates fast secretion of FliK into the periplasm during rod assembly, thereby

preventing a premature substrate specificity switch. However, in the absence of FlhE, slowed-down secretion of FliK during rod assembly might allow time for FliK_C to interact with FlhB and flip the fT3SS-specificity switch before HBB completion, resulting in a non-functional flagellar structure and secretion of FlgM into the periplasm. We therefore measured FliK secretion into the periplasm using a β-lactamase (Bla) reporter fusion to FliK expressed from an arabinose-inducible promoter (P_{araBAD}:*fliK-bla*)[46]. FliK-Bla must be secreted into the periplasm via the fT3SS in order to confer resistance to ampicillin, which can be quantified as the minimal inhibitory concentration (MIC). Expression of FliK-Bla in the Δ*flgI* and Δ*flgH* background resulted in an MIC reduction from 25 μg/ml Ap in presence of FlhE, to 12 and 6.25 μg/ml Ap, respectively, when FlhE was deleted (Fig. S7f). No differences in FliK-Bla secretion were observed for any other rod deletions. However, in a FlhB_{N269A} autocleavage mutant that is unable to undergo the substrate specificity switch, FliK-Bla secretion was restored to WT levels in the Δ*flgI* Δ*flhE* and Δ*flgH* Δ*flhE* mutants. The fact that FliK-Bla secretion is reduced in Δ*flgHI* Δ*flhE* mutants compared to the WT, but restored in the Δ*flgHI* Δ*flhE* FlhB_{N269A} mutant suggests that FlhE is not involved in facilitating FliK secretion during rod assembly. The reduced FliK-Bla secretion levels in the Δ*flgHI* Δ*flhE* background can be explained with the fact that the absence of FlhE results in a premature substrate specificity switch, which prevents further FliK-Bla secretion.

### Live cell labelling reveals assembly of periplasmic flagella in the absence of FlhE
As demonstrated above, filament assembly is necessary for the cell death phenotype in the Δ*flgHI* Δ*flhE* mutant. In order to enable visualization of filament assembly in live cells, we utilized a previously

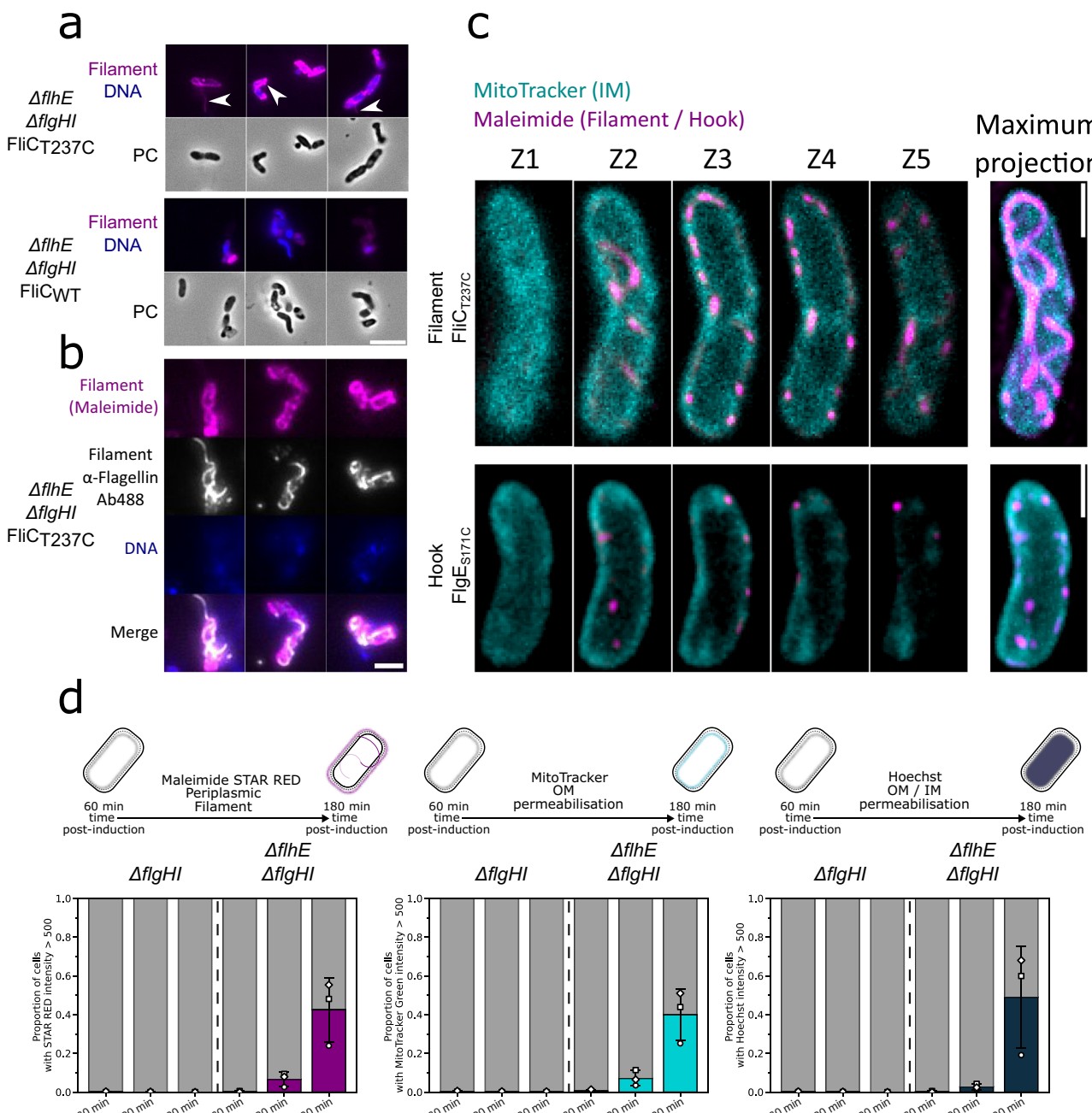

**Fig. 4 | Loss of FlhE in the *ΔflgHI* background cause flagellar filament to assemble into the periplasm. a** Maleimide staining (magenta) of flagellar filament in the *ΔflhE ΔflgHI* using FliC$_{T237C}$ and locked in FliC mode (*Δhin*, FliC-ON). Extracellular filament can be visualized. Additional intracellular structure can be observed and are absent in a *ΔflhE ΔflgHI* using WT FliC without cysteine. DNA staining with DAPI is represented in blue. Scale bar = 5 μm. White arrows indicate the extracellular/intracellular structures observed. **b** Immunostaining (white) of flagellar filament coupled with Maleimide staining (magenta) in *ΔflhE ΔflgHI* FliC$_{T237C}$. Colocalization between immunostaining and maleimide staining confirm the structures observed are flagellar filament. Scale bar = 2 μm. **c** Flagellar filaments are assembling in the cells periplasmic space. 3D STED microscopy of *ΔflhE ΔflgHI* FliC$_{T237C}$ (upper panel) and *ΔflhE ΔflgHI* FlgE$_{S171C}$ (bottom panel) after STAR RED maleimide staining (Magenta) and IM staining with MitoTracker Green (Cyan).

Z-stacks with 80 nm intervals were acquired. Left panel represents pictures at the different Z-position, right picture the maximum projection of the Z-stack acquired. Scale bar = 1 μm. Data shown are from one representative experiment and have been independently replicated in at least one preliminary experiment. **d** Time course epifluorescence microscopy. P$_{tetA}$-flhDC *ΔflhE ΔflgHI* strain was induced with AnTc and aliquoted after 60, 120 and 180 min. Cells were simultaneously stained with Maleimide STAR RED (filament staining, magenta), MitoTracker Green (IM, cyan) and Hoechst (DNA staining, blue). Biological triplicates were performed, and at least 283 cells were analyzed per timepoint. White symbols represent the mean of each biological replicate and error bars indicate the standard deviation. Cells with a mean intensity >500 were considered as positive. Assembly of flagellar filament is causing an increase in IM staining and DNA staining due to increase permeabilization of the OM/IM. Source data are provided as a Source Data file.

described cysteine replacement mutation of flagellin (FliC$_{T237C}$)[44], which allows to label the flagellar filament using maleimide-coupled fluorophores. Upon labelling with maleimide STAR RED, we were able to observe extracellular filaments as expected. Interestingly, however, we were able

some cells exhibited irregular intracellular, filament-like structures. In absence of FliC$_{T237C}$, no filament-like structures were observed (Fig. 4a). To ensure that the observed structures were genuine flagellar filaments, we sequentially labelled FliC$_{T237C}$ with maleimide STAR RED

and anti-FliC antibodies, visualized using Alexa488-coupled secondary antibodies (Fig. 4b). For most cells, we observed intracellular filament-like structures stained with STAR RED and Alexa488-stained flagellar filaments externally (Fig. 4b). However, some cells displayed intracellular double staining, demonstrating that the observed STAR RED-stained filament-like structures were in fact flagellar filaments. We next used super-resolution stimulated emission depletion (STED) microscopy to demonstrate that these filament-like structures extend into the periplasmic space, as they colocalize with the membrane staining dye MitoTracker (Figs. 4c and S8). Similar results were obtained using maleimide STAR RED labelling of a cysteine mutant of the hook (FlgE$_{S171C}$)(Fig. S8). These findings suggest that periplasmic flagellar formation is the primary cause for the observed cell death and cell morphology defect in the absence of FlhE. We then wondered if the change of curvature was caused by the natural helical conformation of the flagellar filament, and if modifying this conformation from helical to straight, curly or coiled would impact the shape of the cells. To do so, we used previously described point mutants of the flagellin FliC, which modify the conformation of the flagellar filaments[47,48]. Interestingly, none of the point mutants prevented the change in the cell curvature, nor the growth defect observed in Δ*flgHI* Δ*flhE* mutant, suggesting that the helical shape of the flagellar filament was not responsible for the observed increase of cell curvature itself (Fig. S9). Due to the formation of periplasmic filaments and defects in morphology and membranes, we did not expect the Δ*flgHI* Δ*flhE* mutant to exhibit motility. However, using phase contrast microscopy, we observed that a few cells with periplasmic filaments displayed motion in liquid medium despite their morphology defects (Supplementary Movie 1). Strikingly, some cells exhibited rotation of the entire cell body, resembling the gyratory motion observed for *Leptospira* (Supplementary Movie 1, right panel)[49].

### Membrane and cytoplasmic co-staining suggest that defects in early PG assembly cause aberrant morphology and cell lysis

To better understand how periplasmic filament assembly affects cell viability, we investigated the effects of the absence of FlhE on OM and IM integrity using simultaneous labelling with maleimide STAR RED (indicating OM permeability by labelling periplasmic filaments), MitoTracker (indicating OM permeability by labelling the IM), and Hoechst (indicating OM and IM permeability by labelling the DNA). Time-course analysis showed that overall fluorescence for all three dyes increased in the Δ*flgHI* Δ*flhE* background but remained constant in the Δ*flgHI* FlhE⁺ background (Fig. 4d). We however noted that the increase in the shape curvature was not correlated to the increase in the fluorescence signal (Fig. S10a). The STAR RED signal and the MitoTracker Green signal correlated with the Hoechst signal, indicating that the presence of periplasmic filaments was associated with increased OM and IM permeabilization (Fig. S10b). These observations suggest that the curvature defect (likely due to impairment in spatial control of PG assembly caused by periplasmic filament assembly) precedes OM permeabilization. Prior studies have shown that cells can survive cell wall defects for several hours in the absence of OM permeabilization, emphasizing a mechanical role for the OM[35] (Fig. 2c). Taken together, our results suggest that the growth of periplasmic filaments, triggered by the absence of FlhE, exerts stress on the cell wall and subsequently compromises the proton gradient across the IM. Together, these perturbations affect the function of the elongasome and divisome, resulting in aberrant cell morphology and leading eventually to cell lysis.

### Electron microscopy reveals shorter rods and the presence of periplasmic hook structures in a PL-rings mutant in the absence of FlhE

In order to validate our STED microscopy observations of periplasmic hook assembly, we purified HBBs from the WT, Δ*flhE* and Δ*flgHI* Δ*flhE*

mutants, and imaged the isolated HBBs using transmission electron microscopy (TEM). As anticipated, HBBs purified from the Δ*flgHI* Δ*flhE* strain displayed a rod and a hook structure, while the PL-rings were absent (Fig. 5a). Quantifications of the length of the rod, hook and entire structure, revealed that rods and the entire HBB structures were on average shorter in the Δ*flgHI* Δ*flhE* mutant compared to WT. Surprisingly, the lengths of the hooks were similar between the mutants, although we expected longer hooks to compensate for the shorter rods structure (Fig. 5b). These observations confirm that hooks can grow in the periplasm in the absence of FlhE in the PL-rings mutant, and suggest a premature switch from rod assembly to hook assembly prior to completion of the rod structure, in the absence of FlhE. Importantly, while most HBB structures of the Δ*flhE* mutant resembled the standard HBB structure containing both P- and L-rings, some either lacked the L-ring (with only the P ring present, Fig. 5c) or were completely ringless (Fig. S11a, b). Similar to the Δ*flgHI* Δ*flhE* mutant, the rods in these 'truncated' HBBs were shorter. However, hook length was also reduced in absence of the PL-rings (Fig. S11c). The reason for this decreased length is unclear. However, these observations demonstrate that some basal bodies in the Δ*flhE* strain were unable to assemble either the L-ring, or both the P- and L-rings, and subsequently switched to premature hook assembly inside the periplasm.

## Discussion

The flagellar structure of Gram-negative bacteria are composed of several thousands of building blocks, traverse multiple layers of the cell envelope, including the IM, PG layer, and OM[3]. The complex pathway associated with flagellum assembly is contingent on a well-orchestrated hierarchy of gene expression and secretion of building blocks, critical to prevent premature secretion of flagellar building blocks, and consequently, wasted biosynthetic resources. Assembly of the rod in *S. enterica* follows the same principle. Once the IM rings are formed, rod subunits are secreted via the fT3SS into the central channel of the growing periplasmic structure and assembled at its tip. Proximal rod subunits (FliE, FlgB/C/F) undergo sequential polymerization up to the PG layer. The muramidase FlgJ assembles on top of the proximal rod and acts as the rod-cap, enabling the polymerization of the distal rod subunits FlgG. Crucially, FlgJ is required to digest the PG layer, a step necessary for the continued elongation of the rod. Upon reaching the OM distal rod polymerization ceases followed by PL-rings completion, forming an aperture that allows the fT3SS structure to continue its growth beyond the cell envelope. Absence of FlgH or the PG ring component FlgI results in the cessation of rod growth at the OM level[3,9].

Among the forty known gene products associated with the construction and regulation of bacterial flagella, the role of FlhE in flagellum assembly is one of the few that remained unclear. Interestingly, we found FlhE to be restricted to a few bacterial phyla (Supplementary Dataset 1). Several lines of evidence suggest that FlhE was recruited to the flagellar system recently. First, FlhE homologs are lacking from many genomes of diverse bacteria that have flagella. Second, they are present in some genomes that do not have any flagellar genes, suggesting an alternative function. Third, *flhE* genes are found in flagellar operons only in three orders of Gammaproteobacteria (which is a recently evolved bacterial clade). Phyletic distribution of FlhE orthologs and their operon locations suggest that the *flhE* gene was recruited to the flagellar operon of a *Burkholderiaceae* ancestor, from where it was horizontally transferred to several other gammaproteobacterial lineages, including *Enterobacteriaceae* (Fig. S1a). This scenario agrees with a recent report that enteric bacteria acquired their entire flagellar and chemotactic systems through horizontal gene transfer from *Burkholderiaceae*[50]. FlhE might have undergone adaptive evolution, acquiring new functions or roles beneficial to those specific bacteria. FlhE also displays a remarkable similarity to the periplasmic protein FlaA from *Leptospira barantonii* (Fig. S1b) and which therefore

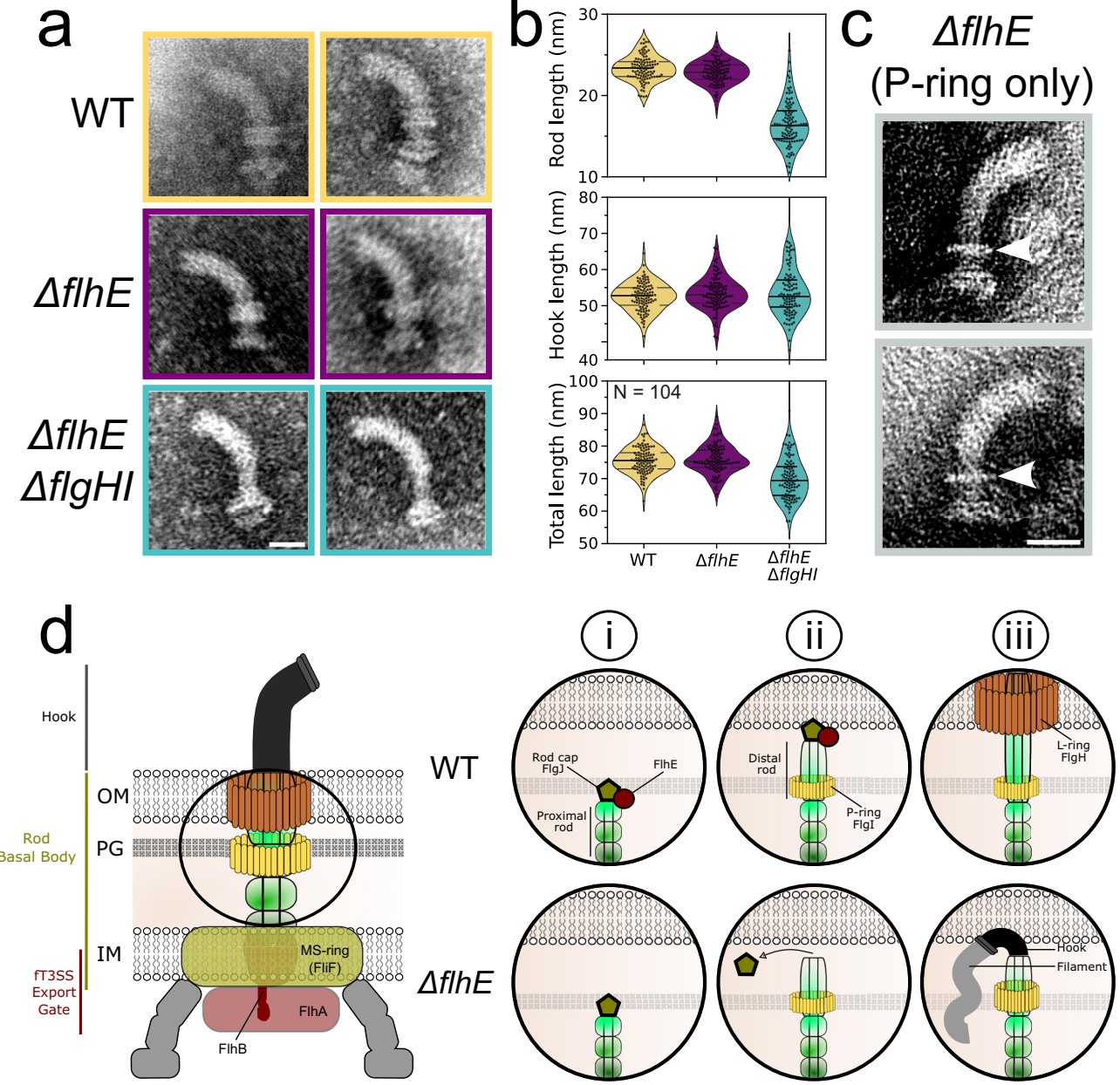

**Fig. 5 | *ΔflhE ΔflgHI* mutant assembles shorter rods than WT. a** Exemplary pictures of HBB purified from WT, *ΔflhE* and *ΔflhE ΔflgHI* strains, observed by TEM. Scale bar = 25 nm. Data shown are from one representative experiment and have been independently replicated in one preliminary experiment. **b** Length of rod, hooks and complete HBB structure from pictures shown in **a**. *ΔflhE ΔflgHI* rods and HBBs total lengths are shorter than WT. Hook length remain similar. Source data are provided as a Source Data file. **c** Purified HBBs in the *ΔflhE* FlgHI⁺ strain show that some HBBs are unable to assemble a L-ring and only displayed a P-ring. White arrows point the assembled P-ring. Scale bar = 25 nm. **d** Model for function of FlhE during rod assembly. (i) The rod cap FlgJ assemble on top of the proximal rod,

acting as a scaffold protein for polymerization of the rod subunits. Upon contact of the PG layer, the muramidase activity of FlgJ degrades the glycan strands and enables penetration of the rod into the PG layer to continue its growth. (ii) Upon contact with the OM, FlhE stabilises the rod cap FlgJ and prevents assembly of hook subunits prior to L-ring assembly. In absence of FlhE, contact with the OM of the rod cap (or prior contact with the PG layer) causes the rod cap to fall of the polymerizing rod structure. (iii) Loss of the rod cap in *ΔflhE* strains enables the assembly of hook subunits, followed by flagellin assembly leading to periplasmic flagellar filament formation.

might have similar functions of controlling some part of the assembly of periplasmic flagella in spirochetes. FlaA isoforms constitute a proteinaceous sheath that covers the periplasmic filament core, composed of FlaB isoforms. FlaA2 was recently observed to interact with glycosylated residues of the flagellin FlaB, contributing extensively to the filament-sheath interfaces[51]. However, no such post-translational processing is known for flagellar proteins in *S. enterica*, and the relation between FlhE and FlaA remains to be investigated.

However, using *S. enterica* as model for flagella assembly in Gram-negative bacteria, our research provides critical insights into the functional role of FlhE. The loss of FlhE triggers an early substrate specificity switch, as reported previously by Hirano et al.[52], and initially suggesting a potential role of FlhE in preventing early secretion of flagellin prior to complete assembly of the HBB. We discovered that the absence of FlhE leads to the formation of periplasmic flagellar filaments, disrupting various cellular processes and explaining this

early substrate specificity switch observed previously. Notably, the deletion of *flhE*, in conjunction with the absence of the PL-rings, results in the formation of shorter rods and abnormal periplasmic growth of hooks and filaments. These aberrant structures, comprising thousands of flagellin subunits and extending up to 10 µm, disrupt key cell wall regulatory complexes, namely the elongasome and divisome. Intriguingly, some filaments in the Δ*flgHI* Δ*flhE* background were observed outside the cell envelope. This suggest that those filaments, initially confined in the periplasm, continue to grow until they are able to pierce the OM, extending into the extracellular space. This phenomenon raises the possibility of similar breaches in the IM. The observed mis-assembly and mis-localization of the divisome and elongasome could therefore stem directly from the periplasmic filament growth affecting IM integrity and compromising the PMF. Ultimately, the compromised integrity of the OM and IM leads to cell lysis, as evidenced by the observed phenotype on culture plates.

Given those observations, we propose a refined model where FlhE functions as a periplasmic chaperone, crucial for stabilizing the interaction between the rod-cap FlgJ and the distal rod during flagellar assembly (Fig. 5d). Specifically, in the presence of FlgI and FlgH, FlhE ensures the stability of the rod cap until the formation of the L-ring, effectively guiding the correct formation of the basal body structure until it extends outside the cell envelope. FlhE ensures the function of the rod cap until formation of the L-ring in the OM, possibly suppressing the replacement of FlgJ by the hook-cap FlgD and thereby preventing the formation of unwanted periplasmic flagellar filaments. Conversely, in the absence of FlhE, the rod-cap might prematurely dissociate from the rod-tip, triggering the assembly of the hook-cap and hooks subunits followed by filament formation within the periplasm. Interestingly, the loss of FlhE displayed a stronger growth defect in the absence of the PL-rings. Our model for the function of FlhE provides an explanation for this observation as prolonged contact of the rod cap with the OM could lead to FlgJ removal, thereby allowing for hook assembly in the periplasm. It is also possible that loss of the rod cap could occur by interactions with the PG layer in the absence of FlhE. This hypothesis is supported by the presence of HBB structures that contained only the P-ring (or no rings) in the Δ*flhE* FlgHI$^+$ strain and by the variability in rod and hook lengths observed in the Δ*flgHI* Δ*flhE* background. The high variability in rod lengths in the Δ*flgHI* Δ*flhE* background suggests that the loss of the rod cap might be influenced by the assembly stage and its interaction with cellular structures. In WT, hook length is determined by the molecular ruler FliK, resulting in a total length of the HBB structure of ~80 nm (~25 nm rod length and ~55 nm hook length). In the case of periplasmic hooks, we therefore expected the hook length to be longer to complement for the shorter length of the rods, and the entire structure to have the same length as the WT. As measured from the TEM HBB images, hook length of Δ*flhE* mutants was indeed similar to WT, yet the length of the entire structure was shorter in the Δ*flgHI* Δ*flhE* mutant. The reason for the decreased total length of the periplasmic HBB structures is unclear. FliK secretion into the periplasm in the Δ*flgHI* Δ*flhE* mutant might be slower or periplasmic FliK might be prone to degradation. The observed growth defects in the Δ*flgHI* Δ*flhE* mutant raise questions about the non-essential nature of FlhE in the presence of FlgHI. Given the fT3SS's optimization for rapid assembly, enabling swift cellular motility, we hypothesize that in the absence of FlhE, the rod may reach the OM and form the OM ring before the rod cap is dislodged. Future biochemical studies to identify protein interactions with FlhE will be crucial to provide new insights into FlhE's function and the rod assembly, and verify the model we propose here.

Finally, we observed that in the Δ*flgHI* Δ*flhE* mutant some cells retained motility despite their morphology defects. Strikingly, these cells displayed a motility motion similar to that observed in spirochaetes that naturally assemble periplasmic filaments. This observation highlights a potential ancestral mechanism for motility and assembly of flagellar filaments, which diverged through evolution and was lost in *Enterobacteriaceae*.

Altogether, our results demonstrate a role for FlhE in optimizing flagellar assembly and preventing ectopic periplasmic flagellar filament formation, which leads to drastic cell morphology defects and lysis. Those observations provide new insights into the assembly of the periplasmic flagellar rod, and exemplify how bacterial flagella are a paradigm of evolutionary adaptation, requiring optimal, precise assembly to achieve a harmonized function between all the components that make up its structure[11,53].

## Methods

### Bacterial strains and growth conditions

Bacterial strains used in this study are derivatives from *Salmonella enterica* serovar Typhimurium strain LT2 and are listed in Tables S1 and S2. Plasmids and primers used in this study are listed in Tables S3 and S4, respectively. Bacterial strains were cultivated aerobically at 37 °C in liquid culture at 180 rpm or on solid media in Lysogeny Broth-Lennox (LB, 10 g/L tryptone, 5 g/L yeast extract, 5 g/L NaCl) liquid medium and supplemented with 12 g/L agar for agar plates. Overnight cultures were prepared by inoculating single colonies in LB medium. Day cultures were prepared by diluting overnight cultures 1:100 and cultures were grown to mid-exponential phase, if not stated otherwise. Antibiotics were added as follows if needed for plasmid maintenance: chloramphenicol 12.5 µg/mL, ampicillin 100 µg/mL, and kanamycin 15 µg/mL or at the indicated concentrations for the killing curve assay and induction of P$_{tetA}$-*flhDC* on plate.

For induction of gene expression from the P$_{araBAD}$, P$_{tetA}$ and P$_{lac}$ promoters, 0.2% arabinose, 100 ng/mL anhydrotetracycline (AnTc) and 250 µM IPTG were used, respectively. Construction of strains by chromosomal recombination using the λ-RED system was performed as described in refs. 54,55.

### Green plate as indicators for Δ*flhE* phenotype assessment

Green plates were prepared with the following components: 7.4 g/L D-glucose, 7.9 g/L tryptone, 1 g/L yeast extract, 4.9 g/L NaCl, 12 g/L agar, 65 mg/L methyl blue, 0.6 g/L alizarin yellow G. Plates were stored at 4 °C. Strains were restreaked from the −80 °C stock on green plate and incubated at 37 °C between 12–16 h. Color of the colonies was visually assessed the next day. For each plate, a WT control (*S. enterica* LT2, TH437 white) and Δ*flhE* control (TH12483/EM11563, blue) were restreaked to confirm the colony color phenotype.

### Swimming motility test in agar

2 µL of overnight culture incubated at 37 °C was inoculated into swimming motility agar plates and incubated at 37 °C for 4–6 h. If required, arabinose was added to the plate prior to the agar for a final concentration of 0.2%. Images were acquired using a Perfection V800 Photo scanner (Epson). The diameter of swimming halos were measured manually using Fiji[56] and values were normalized to the WT.

### Propidium iodide assay

Cells were stained with the Viability/Cytotoxicity Assay Kit for Bacteria Live & Dead Cells (Biotium). Cells were stained according to the manufacturer's specifications. Ethidium Homodimer III (EthD-III) purchased with the kit selectively stains dead bacteria with damaged cell membranes, as a red-fluorescent nucleic acid dye. Briefly, cells were grown overnight in LB at 37 °C and diluted 1:100 in 5 mL LB at 37 °C for 3 h. For the strains carrying a P$_{tetA}$-*flhDC* promoter, AnTc was added in the culture after 1 h of subculture growth at a final concentration of 100 ng/mL. Aliquots of each culture were probed, washed twice with PBS after centrifugation at 2500 × *g* for 2 min. Dyes were added at the final concentration of 2 µM for EthD-III. After 30 min incubation at 37 °C, 300 rpm in the dark, cells were washed as described above and injected in a custom-made flow cell coated with poly-L-Lysine. Custom

made flow-cell chambers were constructed using slides and cover slips treated with 0.1% poly-L-Lysine[57] (Sigma-Aldrich). The slide and cover slip were assembled in the presence of a double layered parafilm as a spacer. In brief, poly-L-Lysine coated coverslips were loaded with 100 μL cells, incubated for ~3 min and then mounted with Fluoroshield with DAPI (Sigma-Aldrich). Slides were observed with an inverted Zeiss Axio Observer Z1 microscope equipped with AxioCam 506 Mono camera and with 49 DAPI, 38 He GFP and 43 DsRed filters (Zeiss). For analysis, a fixed threshold was applied to distinguish dead cells to live cells in all samples for comparison. A damaged cell membrane control was performed for each replicate with cells fixed with 4% PFA prior to labelling with the fluorescent dye.

### Morphology measurements and FliG-mNeonGreen/FlgE$_{S171C}$ staining

Overnight cultures incubated at 37 °C (supplemented with AnTc 100 ng/mL for strains carrying the P$_{tetA}$-flhDC promoter) were diluted 1:10 in 500 μL PBS. Maleimide DyLight™ 550 Maleimide was added to a final concentration of 10 μM and incubated with the cells for 10 min at room temperature with 300 rpm agitation. Cells were centrifugated 2500 × g for 2 min. Supernatant was removed and cells were resuspended in PBS. Washing step was repeated a second time. Cells were applied on a 1% PBS agarose pad and imaged using a Zeiss Axio Observer Z1 inverted microscope equipped with Pln Apo 100x/1.4 Oil Ph3, an Axiocam 506 mono CCD-camera and LED Colibri 7 lightsource (Zeiss). Images were taken using the Zen 3.8 pro software and the filters 90 HE LED (E) and 60 HE CFP + YFP + HcRed shift free (E)(Zeiss). Z-stack images were acquired every 0.4 μm on a 2.4 μm range. GFP and DsRed intensity and exposure were 250 ms 100% and 500 ms 100%, respectively.

### Construction vector MreB-mNeonGreen/mNeonGreen-MinD

Plasmids pEM12582 (MreB-mNeonGreen) and pEM12581 (MinD-mNeonGreen) were constructed as follow. Overlapping PCR is a variant of standard PCR, consisting in assembling several fragments amplified by PCR in one reaction. Sequences of interests were amplified separately with Q5 High-fidelity polymerase, with oligonucleotides with flanking regions homologous to the forward/reverse fragment (primers 5401 to 5406 for *mreB*, 5409 to 5412 for *minD* in Table S4). Fragments were gel-purified using the NucleoSpin® Gel and PCR Clean-up Kit (Macherey-Nagel). PCR products were mixed together in a ratio 1:1 with Q5 PCR mix in absence of primers and incubated in the PCR thermocycler. Initial 10 cycles enable amplifications of the final construct using the PCR products and their flanking regions to polymerise the final fragment length. After 10 cycles, PCR was paused, 0.5 μM of primers amplifying from the 5′-end of first fragment to 3′-end of the last fragment were added. PCR was resumed for 25 additional cycles. Amplification was checked by electrophoresis and correct fragment was purified using the NucleoSpin® Gel and PCR Clean-up Kit (Macherey-Nagel) for subsequent Gibson Assembly. Plasmids were constructed by Gibson Assembly using Gibson Assembly Master Mix or HiFi Master Mix purchased from NEB. The vector was linearized by PCR (primers 5407/5408 for *mreB*, 5413/5414 for *minD* in Table S4) and gel purified after treatment with DpnI (NEB) for 2–4 h at 37 °C to digest methylated plasmid template DNA. The insert was amplified with specific primers containing 20–40 bp overhangs to the linearized vector insertion site. The Gibson Assembly reaction was performed using 0.02–0.05 pmol linearized vector and 0.1–0.2 pmol purified insert. The samples were incubated for 1 h at 50 °C before cleaning (NucleoSpin® PCR Clean-up Kit (MachereyNagel)) and transformation of 3 μL of purified plasmid into electrocompetent *S. enterica* serovar Typhimurium cells prepared freshly. Transformants were spread on selective LB agar, checked by PCR and positive PCR products were sent to sequencing after purification. MinD-mNeonGreen was then integrated in the chromosomal native locus using the λ-RED system.

### FtsZ Z-ring and MreB visualisation by confocal microscopy

Plasmid pXY027-*ftsZ-eGFP* used for visualisation of the Z-ring was a gift from Jie Xiao (Addgene plasmid # 98915)[34]. For time-lapse observation of pXY027-*ftsZ-eGFP*, overnight cultures were diluted 1:100 in 5 mL LB in presence of chloramphenicol but absence of IPTG inducer and let grown at 37 °C until reaching mid-exponential phase. Cells were washed once in LB diluted 1:5 with PBS at 11,000 × g for 2 min and applied on 1% agarose pad in 1:5 LB-PBS. Images were acquired using Nikon inverted Ti-2 Eclipse using Kinetix camera (Photometrics) in 16 bit camera mode. Pictures were taken every 10 min with 5% GFP laser 25 ms and a 0.8 μm range at 0.4 μm interval Z-stack in a 37 °C pre-warmed incubation chamber. Chromosome localisation/repartition was done with an identical workflow as indicated above, except the following: after reaching mid-exponential phase, cells were washed twice in 1:5 LB-PBS. 1 μM Maleimide 560 LIVE (Abberior) was added and incubated 30 min at 37 °C with low agitation (300 rpm) to stain the DNA. Cells were washed again twice and put on agarose pad 1:5 LB-PBS (1% agarose). Cells were then imaged using the Nikon Ti-2 Eclipse inverted microscope with the following settings: CFI Plan Apochromat DM 60x Lambda oil Ph3/1.40 objective (Nikon), Kinetix 16-bit camera mode with 0.8 μm range at 0.4 μm interval Z-stack, exposure 100 ms 20% GFP laser and 100 ms 20% DsRed. Confocal visualisation of FtsZ-eGFP and Maleimide 560 LIVE stained bacterial cells was performed on stationary phase culture on the Abberior STED using the confocal lasers Pulsed Diode Laser PDL-T 488 and Pulsed Diode Laser PDL-T 561 and Z-stack imaging the whole cell with 75–90 nm interval. Plasmids pXY027-*mreB*-mNeonGreen and pXY027-mNeonGreen-*minD* were constructed by overlapping PCR and Gibson Assembly using pXY027-*ftsZ*-eGFP as template. pXY027-*mreB*-mNeonGreen confocal observations were performed on stationary phase culture grown at 37 °C overnight and applied on 1% agarose pad in PBS and Z-stack imaging of the whole cell with 75–90 nm interval.

### Antibiotic susceptibility test

Overnight cultures incubated at 37 °C were diluted 1:50 in 2 mL LB, grown at 37 °C for 8 hours and were diluted to OD$_{600}$ = 0.01. 198 μL of the diluted culture were loaded in 96 well plates. 2 μL of the antibiotic solution at the corresponding concentration were added in corresponding wells after 1 h incubation at 37 °C in Tecan infinite M200 NanoQuant Plate™ (TECAN). Plates were incubated overnight at 37 °C with agitation. OD$_{600}$ quantifications were plotted 8 hours after addition of the antibiotic for each antibiotic concentration. For microscopy observations, overnight cultures were incubated at 37 °C and were the next day 1:100 in 25 mL LB and grown at 37 °C. Until reaching mid-exponential phase, cultures were balanced to OD$_{600}$ = 0.5 and treated with the corresponding antibiotic. Growth was resumed at 37 °C for 1 hour 30 and cells were directly applied on 1% agarose pad. Fraction of dead cells was measured manually with CellCounter plugin from Fiji by counting cells with a loss of contrast indicating a loss of cytoplasm content.

### SYTOX labelling

For SYTOX GREEN labelling to confirm integrity of the OM, overnight cultures of the strains cultivated at 37 °C were diluted 1:100 in 5 mL LB and grown at 37 °C until reaching mid-exponential phase. OD$_{600}$ was measured and balanced to OD$_{600}$ = 0.2 in PBS for each strain. Aliquots were washed once with PBS at 13,000 × g for 1 min before adding SYTOX GREEN (final concentration 50 nM). Cells were incubated 10 min at 37 °C with low agitation at 300 rpm, washed twice with PBS at 13,000 × g for 1 min and concentrated in 100 μL fresh PBS. Cells were spotted on 1% agarose pad diluted with ddH$_2$O. Control of damaged OM was done simultaneously with WT cells incubated for 30 min in 70% ethanol prior to the washes and labelling with SYTOX GREEN. Images were acquired on Nikon Ti-2 Eclipse inverted microscope with a

60x objective, Fusion FT 16-bit mode camera and with GFP laser 5% 50 ms.

## CellAsic ONIX microfluidic experiment

CellASIC ONIX Microfluidic Platform and CellAsic Onix plates B04A (Merck) were used for live-cell observations. WT and Δ*flhE* bacterial cultures were grown separately at 37 °C overnight in LB, supplemented with ampicillin and were diluted 1:100 in the same medium at 37 °C until reaching exponential phase. Cells were backdiluted to $OD_{600} = 0.02$ and mixed in a 1:1 ratio and injected in CellAsic Onix plates B04A. A continuous flow of medium (LB + ampicillin) was injected in the chamber at 1 psi. Pictures were taken every 20 min in 37 °C pre-warmed Zeiss Axio Observer Z1 inverted epifluorescence microscope with a Pln Apo 100x/1.4 Oil Ph3 objective, an Axiocam 506 mono CCD-camera and lightsource HXP R 120 W/45 C VIS (Zeiss). Images were taken using the Zen 2.6 pro software and the following filters: 46 He YFP and CFP ET Filterset (AHF). For both fluorescence, settings used were 5% laser intensity and 50 ms exposure.

## Immunostaining of the filament

Immunostaining of the filament was performed to visualize flagellar filaments in absence of FliC cysteine substitution required for maleimide labelling. Custom made flow-cell chambers were constructed using slides and cover slips treated with 0.1% poly-L-Lysine as described above. Cells were fixed with 4% PFA for 10 min. Every step was performed at room temperature. After that, cells were washed with PBS and blocked with 10% BSA for 10 min. Primary antibody (anti-FliC, rabbit, 1:1000 in 2% BSA, Becton Dickinson & Company) was added and incubated for 1 h. Subsequently, cells were washed and blocked again as described above. Secondary antibody (anti-rabbit Alexa Fluor488, diluted 1:1000 in PBS, Thermo Fischer Scientific) was added and the mix was incubated for another 30 min. Finally, cells were washed twice with PBS and a mounting solution (Fluoroshield with DAPI, Sigma-Aldrich) was added. Fluorescence microscopy was carried out using a Zeiss Axio Observer Z1 inverted epifluorescence microscope with a Pln Apo 100x/1.4 Oil Ph3 objective, with Z-stack every 0.4 μm with a range of 2.4 μm (7 slices).

## Spot-assay

Single colony cultures were performed in 96-well plates (Corning) at 37 °C with 180 rpm agitation for ~16 hours in 200 μL LB in absence of Tc/AnTc inducer. Cultures were supplemented with antibiotic for plasmid maintenance when required. After growth, cultures were serial diluted in 96 well plates up to a $10^{-5}$ dilution. Serial dilutions were always performed in LB in absence of inducer or antibiotic. Serial dilutions were plated on LB plate agar in absence of inducer or supplemented with Tc, Ap and IPTG, according to the experiment performed. Plates were incubated overnight at 37 °C and imaged the next day using a Perfection V800 Photo scanner (Epson).

## *fliC-mneongreen* transcriptional fusion measurement

Overnight cultures were incubated at 37 °C in absence of AnTc inducer and were diluted the next day 1:100 in 5 mL LB at 37 °C. After 1 hour incubation, cultures were supplemented with 100 ng/mL AnTc and incubation was resumed at 37 °C. Samples were probed every 30 min after addition of the inducer for the time course experiment or at the time indicated for the corresponding experiment. Probed samples were centrifuged at 4000 × *g* for 2 min. Supernatant was removed and cells were resuspended in PBS 1X. Cells were applied on 1% agarose pad in PBS. Samples were imaged using a Zeiss Axio Observer Z1 inverted microscope equipped with Pln Apo 100x/1.4 Oil Ph3 objective, an Axiocam 506 mono CCD-camera and lightsource LED Colibri 7 (Zeiss). Images were taken using the Zen 3.8 pro software and the filters 90 HE LED (E), with GFP intensity and exposure of 10% 100 ms.

## FliK-bla measurement

β-lactamase (lacking its sec-secretion signal) was fused to the C-terminus of FliK via λ-red recombination using primers FliK*bla*-fw and FliK*bla*-rev. Deletion alleles of *flhE* and/or rod component genes were introduced using P22 transduction. The resulting strains are listed in Table S2. The *flhB269A* allele was added to the set of strains described in Table S2 using a Tc-dependent selection-counter selection method. The closely linked *STM1911*::Tn*10d*Tc and Δ*flhBAE7670*::FCF alleles were first introduced by P22 transduction selecting for Tc$^R$ and screening for co-transduction to Cm$^R$. These alleles were then replaced by P22 transduction from select donor strains (TH13763 (*flhB7152*$_{N269A}$ *fljB*$^{enx}$ *vh2*) or TH27928 (Δ*flhE7404 flhB9225*$_{N269A}$) by selection for Tc$^S$ on fusaric acid selection media followed by screening for loss of the Cm$^R$ allele. FliK-bla secretion was evaluated on 6 independent colonies for each strain, using minimal inhibitory concentration (MIC) assays as described in Chevance and Hughes 2023[58]. Briefly, varying LB ampicillin (Ap) solutions were prepared using 2-fold serial dilutions from a freshly made LB solution containing 800 μg/mL of Ap. 198 μL of the varying Ap-concentration solutions were loaded onto clear 96-well plates. Independent cultures of bacteria were grown overnight in 1 mL of LB media under aeration at 37 °C. Overnight cultures were diluted 200-fold in PBS and 2 μL of the diluted cultures were transferred into 96-wells containing varying LB-Ap concentration media. The 96-well plates were incubated, in the dark for 18 hr at 37 °C, with shaking (180 rpm). The MIC value is defined as the first Ap concentration for which the cells are not growing. At least 6 biological independent replicate analysis was performed for each bacterial strain.

## Maleimide staining of the filament

Overnight cultures were incubated at 37 °C in absence of AnTc inducer and were diluted the next day 1:100 in 5 mL LB at 37 °C. After 1 hour incubation, cultures were supplemented with 100 ng/mL AnTc and incubation was resumed at 37 °C. Samples were probed 120 min after addition of the inducer. Probed samples were centrifuged at 4000 × *g* for 2 min, resuspended in PBS supplemented with 10 μM Maleimide STAR RED (Abberior). Samples were incubated 10 min at 37 °C with 300 rpm agitation, washed once with PBS as indicated above, and injected in a custom-made flow-cell chambers coated with 0.1% poly-L-Lysine as described above. Cells were washed twice with PBS and mounted with Fluoroshield with DAPI (Sigma-Aldrich). For dual-staining of FliC using maleimide and anti-FliC antibodies, maleimide staining was performed as described above, and immunostaining as described in the corresponding section. Samples were imaged using a Zeiss Axio Observer Z1 inverted microscope equipped with Pln Apo 100x/1.4 Oil Ph3 objective, an Axiocam 506 mono CCD-camera and lightsource LED Colibri 7 (Zeiss). Images were taken using the Zen 3.8 pro software and a filter 90 HE LED (E).

For STED imaging, the cells were prepared as described above. After incubation with AnTc and washing with PBS, 10 μM maleimide STAR RED (Abberior) and 1 μg/mL MitoTracker Green were added to the samples. Cells were incubated as above, washed twice with PBS and applied on a 1% agarose pad in PBS. Coverslips were cleaned prior with 70% ethanol and plasma oven treatment (1 min on both sides of the coverslips). Samples were imaged with Abberior STED Facility Line equipped with a UPLSAPO 100x oil objective, NA 1.4, a confocal Pulsed Diode Laser PDL-T 488 (10% power, line accumulation 1) for imaging the MitoTracker Green and a Pulsed Diode Laser PDL-T 640 (power 12%, line accumulation 3) and STED laser-40-3000-775-B1R (power 25%, line accumulation 3) for imaging maleimide STAR RED. The pinhole size was set to 0.8–1.0 AU, the pixel dwell time to 5 μs and pixel size to 40 × 40 × 80 nm (xyz). Z-stack images interval was set to 80 nm. A 3D-STED resolution module (Abberior) was used. Images were further processed with Imaris. Fluorescence intensity for the filament staining was deconvolved with Imaris software (Oxford Instruments)

and adjusted in intensity value. Fluorescence intensities for IM staining were adjusted individually to see the contour of the cells.

## Triple membrane labelling

Overnight cultures were incubated at 37 °C in absence of AnTc inducer and were diluted the next day 1:100 in 5 mL LB at 37 °C. After 1 hour incubation, cultures were supplemented with 100 ng/mL AnTc and incubation was resumed at 37 °C. Samples were probed every 60 min after addition of the inducer for the time course experiment. 250 μL probed samples were supplemented simultaneously with 10 μM Maleimide STAR RED (Abberior), 1 μg/mL MitoTracker™ Green FM (ThermoFischer) and 40 μg/mL Hoechst 33342 (Biozol). Samples were incubated 10 min at 37 °C with 300 rpm agitation, washed twice with PBS after centrifugation at 4000 × $g$ for 2 min. Cells were applied on 1% agarose pad in PBS. Samples were imaged using a Zeiss Axio Observer Z1 inverted microscope equipped with Pln Apo 100x/ 1.4 Oil Ph3, an Axiocam 506 mono CCD-camera and lightsource LED Colibri 7 (Zeiss). Images were taken using the Zen 3.8 pro software and the filters 90 HE LED (E), with the following intensity and exposure: STAR RED 10% 100 ms, MitoTracker 5% 50 ms (GFP), Hoechst 5% 30 ms (DAPI).

## ZapA/MinD TIRF measurement

Strains containing chromosomal fusion ZapA-mNeonGreen and MinD-mNeonGreen (N-terminal) at the native locus were grown as indicated above for the triple membrane labelling. Every 60 min after addition of AnTc inducer, samples were probed, washed once in PBS supplemented with 0.2% glucose (4000 × $g$ for 2 min) and applied on a 1.5% agarose pad in PBS supplemented with 0.2% glucose. Cells were then imaged using the Nikon Ti-2 Eclipse inverted microscope with a camera Fusion BT (Hamamatsu), a CFI SR HP Apochromat TIRF 100xC Oil with N.A 1.49, QuadBand filter (F57-407 Quad TIRF ZET405/488/561/ 640 TIRF Quad Laser Rejection) and an ILAS 2 TIRF Module (Nikon) equipped with laser line 488 (200 mW). For ZapA-mNeonGreen, pictures were acquired with 488 nm laser (100% intensity 30 ms) with a Z-stack of 1.6 μm range every 0.4 μm. For MinD-mNeonGreen, pictures were acquired with 488 nm laser (25% intensity 100 ms) and a time-lapse acquisition every 5 seconds for 2 minutes. A perfect focus system (Nikon) was used to maintain the focus. The medial profile of fluorescence intensity for ZapA and demographs for MinD were done with MicrobeJ[59].

## HBB purification

HBB with attached filaments were purified as follows. ONC culture grown at 37 °C were diluted 1:100 in a 500 mL LB flask. After 1 h at 37 °C, cultures were induced with 100 ng/mL AnTc and incubation was resumed for 3 additional hours. Cultures were centrifuged at 8000 × $g$ for 10 min at 4 °C and re-suspended in ice-cold sucrose solution (0.5 M sucrose, 0.1 M Tris-HCl, pH 8). 3 mL 0.1 M EDTA followed by 3 mL 10 mg/mL lysozyme were added drop wise into the samples and incubated at 4 °C for 30 min followed by 30 min at room temperature. Next, 3 mL 10% Triton X-100 and 3 mL 0.1 M MgSO$_4$ were added followed by incubation as described in previous step. After lysis, 3 mL 0.1 M EDTA, pH 11 was added at room temperature and samples were centrifuged at 17,000 × $g$ at 4 °C for 20 min. The supernatant was adjusted with 5 M NaOH to pH 11 and pelleted again with the same conditions. HBB with attached filaments were subsequently collected by ultracentrifugation at 100,000 × $g$ at 4 °C for 1 h. The pellet was re-suspended in an ice-cold pH 11 buffer (10% sucrose, 0.1 M KCl, 0.1% Triton X-100, pH 11) and centrifuged at 100,000 × $g$ at 4 °C for 1 h. Purified flagellar filaments were washed in TET buffer (10 mM Tris-HCl, 5 mM EDTA, pH 8, 0.1 % Triton X-100) and incubated overnight at 4 °C. Samples were centrifugated again the next day at 100,000 × $g$ at 4 °C for 1 h and resuspended in ice-cold pH 2.5 buffer for filament depolymerization (50 mM glycine, 0.1% Triton X-100, pH 2.5). Samples were

incubated 30 min at room temperature and centrifugated at 100,000 × $g$ at 4 °C for 1 h, air-dried for 10 min and re-suspended in 100 μL TET buffer.

## Imaging of the HBB by TEM

Aliquots of HBB samples were applied to freshly glow discharged carbon-film-coated copper grids and allowed to adsorb for 10 minutes. After three washes with distilled water, the grids were contrasted with 4% phospho-tungstic-acid/1% trehalose, touched on filter paper and air-dried. The grids were analyzed in a Leo 912AB transmission electron microscope (Zeiss) at 120 kV acceleration voltage. Micrograph-mosaics were scanned using a bottom mount Cantega digital camera (SIS) with ImageSP software from TRS (Tröndle).

## Bioinformatics analysis of *flhE*

Taxonomy tree and information for phyletic distribution were retrieved from Genome Taxonomy Database (GTDB, Release 95.0)[21]. Protein sequences of representative sets of bacteria were downloaded from the NCBI RefSeq, NCBI nr and GTDB databases. Retrieved data-sets were scanned with the FlhE Hidden Markov Model (HMM) profile (Pfam PF06366) with the E-value threshold of 1. To identify additional sequences overlooked by the model, several sequences from different phylogenetic lineages were used as queries for PSI-BLAST searches against the dataset. The presence of the FlhE domain in all collected sequences was verified using HHpred[60]. Gene neighborhoods were identified using TREND[61] and MiST4[62] databases.

## Reporting summary

Further information on research design is available in the Nature Portfolio Reporting Summary linked to this article.

## Data availability

All data supporting the findings of this study are available within the paper and its Supplementary Information. FlhE distribution across the bacterial tree of life is available as Supplementary Dataset S1. The protein structures from other publications referenced in this paper are accessible under the PDB accession codes 4QXL (FlhE) and AlphaFold model A0A5F2BH85_9LEPT (FlaA). Jupyter notebooks used to plot the graphs are available at: https://github.com/SalmoLab/FlhE_NatComm2024.git Source data are provided with this paper.

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

## Acknowledgements
We thank members of the Erhardt laboratory for helpful discussions. This work was supported by the European Research Council (ERC) under the European Union's Horizon 2020 research and innovation program (agreement no. 864971) to M.E. and by an NIH Grant R35GM131760 to I.B.Z. and NIH PHS Grant GM056141 to K.T.H.

## Author contributions
M.H. and M.E. designed the project. M.H. performed the experiments. E.P.A. and I.B.Z. performed and analyzed the evolutionary model. C.G. performed the TEM grids preparations and observation of the purified HBB samples. F.F.V.C. and K.T.H. created the bacterial strains used for FliK-Bla secretion assay and performed the assay. M.H. wrote the first draft with contribution from E.P.A. M.H. and M.E. wrote the final draft of the manuscript and all of the authors commented on manuscript draft.

## Funding

## Competing interests
The authors declare no competing interests.
