## [Peer Review File · Nature Communications]

FliH functions as a chaperone to prevent formation of periplasmic flagella in Gram-negative bacteriaREVIEWER COMMENTS

Reviewer #1 (Remarks to the Author):

FliH is a flagellar periplasmic protein secreted by the Sec translocon. Loss of FliH function does not affect motility significantly but results in cell lysis. Furthermore, loss of FliH function allows the flagellum-specific protein export apparatus to switch its substrate specificity from early to late substrates in the LP ring mutant background. However, it remains a mystery how FliH is involved in the process of flagellar morphogenesis. Here, Halte et al. provide direct evidence that loss of FliH function results in the formation of periplasmic flagella that leads to a loss of standard cell morphology and cell lysis and propose that FliH functions as a periplasmic chaperone to control rod assembly in the periplasmic space to interfere with periplasmic flagellar formation. All experiments are carefully and well designed and a set of data is clearly presented and well documented. This research article would be of interest to general readership, providing significant advancements in our knowledge on the well-coordinated flagellar assembly process. This reviewer likes such an elegant and beautiful work.

Comments and questions:

1. Kubori et al. have reported that L-ring mutants produce the extended candlestick structure with the proximal part of the hook attached, indicating that the proximal part of the hook is formed on the distal end of the rod even in the absence of the L-ring. This suggests that even though the FlgD cap is at the tip of the hook (Ohnishi et al. *J. Bacteriol.* 176, 2272 – 2281), the hook stops growing unless L-ring forms around the distal rod. In contrast, the authors showed that the hook continues to grow until its length reaches about 55 nm even in the absence of the LP-ring complex. Furthermore, loss of FliH function inhibited LP-ring formation around the rod to a significant degree, indicating that FliH is required for efficient and proper LP-ring formation rather than rod formation. Therefore, this reviewer supposes that FliH coordinates LP ring formation with hook assembly at the tip of the rod rather than controls rod assembly.

2. Hirano et al. have reported that the flgJ mutant lacking the muramidase domain produces basal bodies lacking only the L-ring named candlesticks (*J. Mol. Biol.* 312, 359–369). This suggests that the muramidase activity of FlgJ is required for efficient L-ring formation around the rod. Because the authors showed that loss of FliH function significantly affects L-ring formation around the rod in a way similar to the flgJ mutant lacking the muramidase domain, there is the possibility that FliH controls the muramidase activity of FlgJ during rod formation. In contrast to this flgJ mutant, the flhE null mutant produces the hook structure at the rod tip. Because the L-ring proteins are embedded within the outer membrane to form the L-ring structure around the rod, the rod does not reach the outer membrane in the flhE mutant background, allowing the hook protein to polymerize the hook structure with the help of the hook cap within the periplasmic space. Therefore, FliH would suppress the replacement of the FlgJ cap by the FlgD cap at the rod tip, thereby interfering with the periplasmic flagellar formation.

3. P21, lines 592–624: Did the authors have biochemical evidence that FliH interacts with FlgJ, which acts not only as the rod cap to promote rod assembly but also as a flagellum-specific muramidase that digests the peptidoglycan layer for rod penetration? If not, the authors should tone down their claim in the Discussion.

4. Figure 3d: This reviewer is wondering why the flagellum can extend outside the cell body despite the absence of the LP-ring?

5. P13, line 356: The authors should cite the following three papers in addition to reference No.21.

Bange, G., Kümmerer, N., Engel, C., Bozkurt, G., Wild, K., and Sinning, I. (2010). FliH provides the adaptor for coordinated delivery of late flagella building blocks to the type III secretion system. *Proc. Natl. Acad. Sci. U.S.A.* 107, 11295–11300. doi: 10.1073/pnas.1001383107

Kinoshita, M., Hara, N., Imada, K., Namba, K., and Minamino, T. (2013). Interactions of bacterial chaperone-substrate complexes with FliH contribute to coordinating assembly of the flagellar filament. *Mol. Microbiol.* 90, 1249–1261. doi: 10.1111/mmi.12430

Furukawa, Y., Inoue, Y., Sakaguchi, A., Mori, Y., Fukumura, T., Miyata, T., et al. (2016). Structural stability of flagellin subunit affects the rate of flagellin export in the absence of FliS chaperone. *Mol. Microbiol.* 102, 405–416. doi: 10.1111/mmi.13469

Reviewer #2 (Remarks to the Author):

The bacterial flagellum is an elaborate rotary machine, which is used by the cells for swimming or swarming motility. Normally, a flagellum consists of the extracellular flagellar filament, which is connected via the hook to the cell envelope-embedded basal body, which houses the rotary motor and the export apparatus. Flagellar assembly occurs from inside out, requires some 20 different building blocks at vastly different stoichiometries and is highly regulated at multiple levels. In the paradigm flagellar system of *Salmonella*, the role of almost all proteins directly related to flagellar assembly and regulation is known, with the exception of FlhE. FlhE is a periplasmic protein that is transported via the Sec system. Previous work by another group had demonstrated that loss of FlhE does not affect flagellation and motility but, interestingly, causes infrequent apparent cell lysis. In this work, Halte and colleagues aimed at clarifying the role of FlhE and the cause of the unusual phenotype.

The work starts with a taxonomic analysis to identify the distribution of FlhE within flagellated bacterial species. Homologs were found in several phyla and are present in (only) three orders of the gammaproteobacteria, among them the Enterobacteriales including *Salmonella*. As the next step, the authors validated the findings of the previous work and now also included flagellar mutants that are affected in specific steps of flagellar assembly. In brief, the cell lysis phenotype upon loss of FlhE was suppressed in mutants affected in the flagellar type III secretion system or secretion itself. Motor function and chemotaxis were not apparently involved. In contrast, deletion of the building blocks forming the P-ring or the L-ring strongly exacerbated the cell lysis phenotype. It was also noted that in the absence of FlhE the cells tend to show an aberrant curved morphology, which increased upon stronger expression of the full flagellar apparatus. By a truly impressive array of microscopy approaches the authors demonstrate that the cell machinery for correct placement of the cell division site and PG integration into the nascent cell wall was mislocated in these cells. Most intriguingly, the authors demonstrate by microscopy approaches that loss of FlhE in PL-ring mutants frequently result in formation of flagella which fail to breach the outer membrane, but instead assemble within the periplasm. The formation of periplasmic flagella requires assembly of the flagellar rod (the structure that passes through the periplasm) and the hook. However, as shown by electron microscopy on purified hook-basal body complexes, rod and hook are shorter as in the wild type. The data are consistent with a model where FlhE acts as a chaperone for the rod, e.g. by supporting the rod-rod cap (FlgJ) interaction. In its absence, the nascent flagellar structure may fail to penetrate cell wall and outer membrane. As a result, the flagellar filament forms in the periplasm, thereby interfering with the spatial regulation of cell growth and division site placement and leading to proton leakage.

The authors performed a truly impressive body of work on a huge number of mutants, the quality of the imaging is outstanding. Displacement and of the cellular growth and division machinery by the formation of periplasmic flagella is demonstrated as well as microscopy can do. I am fully convinced that the data presented by the authors justify the model that is finally presented. I wonder if protein-protein interactions, e.g. between proteins of the rod subunit, the rod cap protein, and FlhE have been addressed, which could make the final model more convincing. At least I cannot see it in the manuscript. Even without this, the authors provide a huge step forward into understanding the role of FlhE. Intriguing here is also the structural homology between *Salmonella* FlhE and FlaA of spirochetes, which also form periplasmic flagella.

Please find below a list of issues the author should or may consider:

Table S1 occurs twice, once as a list of the bacteria with FlhE and again as a list of strains used in this study.

47, it would be great to have a reference here for the order of flagella assembly

113, may add ,at the sequence level'

155, it would be nice to have some data here (% of cells) or a reference to a figure where this is shown.

169, is there data for this? And what is meant by a growth defect on plates? Less colonies? Smaller colonies? Blue colonies on GP agar?

171, the sentence should be rephrased, like this it reads as cell lysis only occurs on GP agar.

177, 'lacks an assembled filament' or 'assembled filaments'

181, 'does not assemble filaments'

Fig. 1C, below

Fig. 2A, the two right panels, are those just two different examples of the different phenotype that are observed? Please specify briefly in the caption.

251/Fig. S5c, the cell death after addition of kanamycin also appears to double. Any comments on this?

Figure S5d, the panels are REALLY small, and it is hard to make out what is going on.

272, I would suggest to phrase this a little more carefully (these observations indicate that loss of FlhE causes a defect in the spatial control of PG synthesis) as PG synthesis in general still seems to function.

295, the flgI and flhH phenotypes appear a little bit out of nowhere here, I expected them earlier (e.g. Fig. S2).

318, class III genes?

330, there are only extracellular filaments shown, maybe mention that the staining procedure would not target periplasmic flagella. Otherwise, I would expect to have an image. Or refer to later.

362, after the comma, it's not clear which strain is meant.

388, add 'mutants' in front of 'compared'

426/Fig. 4a, the image is rather small although the result shown is quite important. Maybe it can be enlarged.

439, the different helical forms of the Salmonella flagellum are rather complicated matter that come out of nowhere here. Please add two sentences for explanation and a reference.

484, rather 'spatial control of PG assembly'; see above

539, as mentioned before, are there any interactions shown? Rod subunits, rod tip (FlgJ), FlhE? Has this been tried?

REVIEWER COMMENTS

Reviewer #1 (Remarks to the Author):

FlhE is a flagellar periplasmic protein secreted by the Sec translocon. Loss of FlhE function does not affect motility significantly but results in cell lysis. Furthermore, loss of FlhE function allows the flagellum-specific protein export apparatus to switch its substrate specificity from early to late substrates in the LP ring mutant background. However, it remains a mystery how FlhE is involved in the process of flagellar morphogenesis. Here, Halte et al. provide direct evidence that loss of FlhE function results in the formation of periplasmic flagella that leads to a loss of standard cell morphology and cell lysis and propose that FlhE functions as a periplasmic chaperone to control rod assembly in the periplasmic space to interfere with periplasmic flagellar formation. All experiments are carefully and well designed and a set of data is clearly presented and well documented. This research article would be interest to general readership, providing significant advancements in our knowledge on the well-coordinated flagellar assembly process. This reviewer likes such an elegant and beautiful work.

RE: We are grateful to the reviewer for the positive feedback!

Comments and questions:

1. Kubori et al. have reported that L-ring mutants produce the extended candlestick structure with the proximal part of the hook attached, indicating that the proximal part of the hook is formed on the distal end of the rod even in the absence of the L-ring. This suggests that even though the FlgD cap is at the tip of the hook (Ohnishi et al. J. Bacteriol. 176, 2272 – 2281), the hook stop growing unless L-ring forms around the distal rod. In contrast, the authors showed that the hook continue to grow until its length reaches about 55 nm even in the absence of the LP-ring complex. Furthermore, loss of FlhE function inhibited LP-ring formation around the rod to a significant degree, indicating that FlhE is required for efficient and proper LP-ring formation rather than rod formation. Therefore, this reviewer supposes that FlhE coordinates LP ring formation with hook assembly at the tip of the rod rather than controls rod assembly.

RE: This is an interesting point. Although there has been no chaperone for FlgH and L-ring formation identified so far, it is known that FlgI and P-ring formation requires FlgA as chaperone. In the *LflhE* FlgHI⁺ mutant strain, we observed that after HBB purification some of the HBB were displaying either only a P-ring, or no rings at all, suggesting that the switch from rod to hook assembly can occur independently of the presence of the P- and L-ring. We also note a strong variation of the length of the rod in both *LflhE* *LflgHI* and *LflhE* FlgHI⁺ lacking L-ring and PL-ring. This suggest that rod assembly can be interrupted at different lengths, upon which the hook subunits start to polymerise on top of the rod. Based on those two observations, we concluded that FlhE was involved in the control of rod assembly rather than the LP-ring assembly – similar conclusion as the reviewer summarised in the following comment below.

2. Hirano et al. have reported that the flgJ mutant lacking the muramidase domain produces basal bodies lacking only the L-ring named candlesticks (J. Mo. Biol. 312, 359–369). This suggests that the muramidase activity of FlgJ is required for efficient L-ring formation around the rod. Because the authors showed that loss of FlhE function significantly affect L-ring formation around the rod in a way similar to the flgJ mutant lacking the muramidase domain, there is the possibility that FlhE controls the muramidase activity of FlgJ during rod formation. In contrast to this flgJ mutant, the flhE null mutant produces the hook structure at the rod tip. Because the L-ring proteins are embedded within the outer membrane to form the L-ring structure around the rod, the rod does not reach the outer membrane in the flhE mutant background, allowing the hook protein to polymerize the hook structure with the help of the hook cap within the periplasmic space. Therefore, FlhE would suppress the replacement of the FlgJ cap by the FlgD cap at the rod tip, thereby interfering with the periplasmic flagellar formation.

RE: We agree with this reviewer that this model is at that time the most fitting to our observation. For improved clarity in the Discussion, we added text mentioning the possibility that FlhE suppresses rod cap FlgJ replacement with the hook-cap FlgD at line 608-612: "FlhE ensures the function of the rod cap until formation of the L-ring in the OM, possibly suppressing the replacement of FlgJ by the hook-cap FlgD and thereby preventing the formation of unwanted periplasmic flagellar filaments. Conversely, in the absence of FlhE, the

rod-cap might prematurely dissociate from the rod-tip, triggering the assembly of the hook-cap and hooks followed by filament formation within the periplasm.”

3. P21, lines 592–624: Did the authors have biochemical evidence that FlhE interacts with FlgJ, which acts not only as the rod cap to promote rod assembly but also as a flagellum-specific muramidase that digests the peptidoglycan layer for rod penetration? If not, the authors should tone down their claim in the Discussion.

RE: We have not yet performed biochemical experiments to confirm a putative interaction between the rod-cap and FlhE. Confirming such an interaction biochemically might be technically challenging as the interaction might be transient. This possibility is supported by several lines of evidence in the literature. Importantly, while FlhE was detected in Hook-Basal-Body purification by Lee et al (PMID: **22435757**), demonstrating that FlhE is part of the HBB structure, the recently published complete basal body structure (of a $\Delta flgE$ mutant) obtained by cryo-EM by Johnson *et al*, (PMID: **33931760**) did not contain any density that could be attributed to FlhE. This therefore suggests that any FlhE interaction with the basal body is transient, which fit with the model we are proposing in this manuscript. The absence of density attributed for FlhE in a hook-deficient strain would also agree with our model, as a $\Delta flgE$ mutant assembles the hook-cap FlgD on top of the rod, replacing the rod cap FlgJ. Therefore, interactions between FlhE and FlgJ would likely be missed from the structure obtained. We therefore agree with the reviewer that biochemical experiments investigating this putative interaction between FlhE and the rod-cap should be a focus of future studies.

We added the following sentence in the Discussion in Line 635: “Future biochemical studies to identify protein interactions with FlhE will be crucial to provide new insights into FlhE’s function and the rod assembly, and verify the model we propose here.”

4. Figure 3d: This reviewer is wondering why the flagellum can extend outside the cell body despite the absence of the LP-ring?

RE: This is a good point and was an imprecise description on our side. To clarify, we now make the following statement in Line 594 to discuss this aspect: “Intriguingly, some filaments in the $\Delta flgHI \Delta flhE$ background were observed outside the cell envelope. This suggest that those filaments, initially confined in the periplasm, continue to grow until they are able to pierce the OM, extending into the extracellular space.”

5. P13, line 356: The authors should cite the following three papers in addition to reference No.21.

Bange, G., Kümmerer, N., Engel, C., Bozkurt, G., Wild, K., and Sinning, I. (2010). FlhA provides the adaptor for coordinated delivery of late flagella building blocks to the type III secretion system. *Proc. Natl. Acad. Sci. U.S.A.* 107, 11295–11300. doi: 10.1073/pnas.1001383107

Kinoshita, M., Hara, N., Imada, K., Namba, K., and Minamino, T. (2013). Interactions of bacterial chaperone-substrate complexes with FlhA contribute to coordinating assembly of the flagellar filament. *Mol. Microbiol.* 90, 1249–1261. doi: 10.1111/mmi.12430

Furukawa, Y., Inoue, Y., Sakaguchi, A., Mori, Y., Fukumura, T., Miyata, T., et al. (2016). Structural stability of flagellin subunit affects the rate of flagellin export in the absence of FlhS chaperone. *Mol. Microbiol.* 102, 405–416. doi: 10.1111/mmi.13469

RE: Thank you. We added the suggested references.

Reviewer #2 (Remarks to the Author):

The bacterial flagellum is an elaborate rotary machine, which is used by the cells for swimming or swarming motility. Normally, a flagellum consists of the extracellular flagellar filament, which is connected via the hook to the cell envelope-embedded basal body, which houses the rotary motor and the export apparatus. Flagellar assembly occurs from inside out, requires some 20 different building blocks at vastly different stoichiometries and is highly regulated at multiple levels. In the paradigm flagellar system of *Salmonella*, the role of almost all proteins directly related to flagellar assembly and regulation is known, with the exception of FlhE. FlhE is a periplasmic protein that is transported via the Sec system. Previous work by another group had demonstrated that loss of FlhE does not affect flagellation and motility but, interestingly, causes infrequent apparent cell lysis. In this work, Halte and colleagues aimed at clarifying the role of FlhE and the cause of the unusual phenotype.

The work starts with a taxonomic analysis to identify the distribution of FlhE within flagellated bacterial species. Homologs were found in several phyla and are present in (only) three orders of the gammaproteobacteria, among them the Enterobacteriales including *Salmonella*. As the next step, the authors validated the findings of the previous work and now also included flagellar mutants that are affected in specific steps of flagellar assembly. In brief, the cell lysis phenotype upon loss of FlhE was suppressed in mutants affected in the flagellar type III secretion system or secretion itself. Motor function and chemotaxis were not apparently involved. In contrast, deletion of the building blocks forming the P-ring or the L-ring strongly exacerbated the cell lysis phenotype. It was also noted that in the absence of FlhE the cells tend to show an aberrant curved morphology, which increased upon stronger expression of the full flagellar apparatus. By a truly impressive array of microscopy approaches the authors demonstrate that the cell machinery for correct placement of the cell division site and PG integration into the nascent cell wall was mislocated in these cells. Most intriguingly, the authors demonstrate by microscopy approaches that loss of FlhE in PL-ring mutants frequently result in formation of flagella which fail to breach the outer membrane, but instead assemble within the periplasm. The formation of periplasmic flagella requires assembly of the flagellar rod (the structure that passes through the periplasm) and the hook. However, as shown by electron microscopy on purified hook-basal body complexes, rod and hook are shorter as in the wild type. The data are consistent with a model where FlhE acts as a chaperone for the rod, e.g. by supporting the rod-rod cap (FlgJ) interaction. In its absence, the nascent flagellar structure may fail to penetrate cell wall and outer membrane. As a result, the flagellar filament forms in the periplasm, thereby interfering with the spatial regulation of cell growth and division site placement and leading to proton leakage.

The authors performed a truly impressive body of work on a huge number of mutants, the quality of the imaging is outstanding. Displacement and of the cellular growth and division machinery by the formation of periplasmic flagella is demonstrated as well as microscopy can do. I am fully convinced that the data presented by the authors justify the model that is finally presented. I wonder if protein-protein interactions, e.g. between proteins of the rod subunit, the rod cap protein, and FlhE have been addressed, which could make the final model more convincing. At least I cannot see it in the manuscript. Even without this, the authors provide a huge step forward into understanding the role of FlhE. Intriguing here is also the structural homology between *Salmonella* FlhE and FlaA of spirochetes, which also form periplasmic flagella.

RE: Thank you for the positive feedback on our manuscript!

Please find below a list of issues the author should or may consider:

Table S1 occurs twice, once as a list of the bacteria with FlhE and again as a list of strains used in this study.

RE: Thank you for noticing this, we corrected the order of the tables in the Supplementary Information file.

47, it would be great to have a reference here for the order of flagella assembly

RE: We added the following references at line 48: PMID: **37260402** / PMID: **33572887**.

113, may add ,at the sequence level'

RE: We modified the sentence as suggested.

155, it would be nice to have some data here (% of cells) or a reference to a figure where this is shown.

RE: We added a quantification of the cells at line 154: "However, a subset of cells (~3% of the population) from a 16-hour stationary phase culture showed a loss of phase contrast, indicative of a loss of the cytoplasm."

169, is there data for this? And what is meant by a growth defect on plates? Less colonies? Smaller colonies? Blue colonies on GP agar?

RE: We modified the sentence to make the statement clearer at line 169: "The $P_{synC-flhDC} \Delta flhE$ strain also exhibited slower growth in liquid culture and smaller colonies compared to the $P_{synC-flhDC}$ strain on standard LB agar plates.". We decided not to include a quantification of the growth, as we believe the focus of this paragraph should be on the cell shape defect.

171, the sentence should be rephrased, like this it reads as cell lysis only occurs on GP agar.

RE: We modified the sentence to make the statement clearer at line 173: "however, overexpression of FlhDC in the *LflhE LfliS* background caused the blue phenotype on GP (Fig. S2b)."

177, ,lacks an assembled filament' or ,assembled filaments'

RE: We modified the sentence as suggested.

181, ,does not assemble filaments'

RE: We modified the sentence as suggested.

Fig. 1C, below

RE: We added the following sentence in the legend to make the Fig. 1c clearer: "Schematics below the microscopy images display the flagellation pattern of the WT and P_{synC} ."

Fig. 2A, the two right panels, are those just two different examples of the different phenotype that are observed? Please specify briefly in the caption.

RE: We added the following sentence in the legend of Fig. 2a: "Two distinct cells with ectopic localisation of FtsZ are displayed for $P_{synC-flhDC} LflhE$ background."

251/Fig. S5c, the cell death after addition of kanamycin also appears to double. Any comments on this?

RE: As we observed that the *LflhE* mutant display an increased membrane permeability due to the formation of periplasmic flagellar filament (Fig. 4d), we hypothesized that the compromised membrane integrity was resulting in an increased uptake of kanamycin antibiotic during the assay, leading to this increased cell death in the *LflhE* mutant compared to the WT strain. We however note that PG-targeting antibiotics (cephalexin / ampicillin) have a stronger effect on the *LflhE* mutant.

Figure S5d, the panels are REALLY small, and it is hard to make out what is going on.

RE: We re-arranged the figure, which enabled us to enlarge Figure S5d.

272, I would suggest to phrase this a little more carefully (these observations indicate that loss of FlhE causes a defect in the spatial control of PG synthesis) as PG synthesis in general still seems to function.

RE: We modified the sentence as suggested: "Taken together, these observations demonstrate that loss of FlhE causes a defect in the spatial control of PG synthesis leading to an aberrant cell shape and eventually sets the path for cell lysis."

295, the flgI and flhH phenotypes appear a little bit out of nowhere here, I expected them earlier (e.g. Fig. S2).

RE: We understand the reviewer raising the point of the first mention of the *flgI* and *flgH* mutants in the manuscript. After careful consideration, we decided to maintain the structure as it is at the moment. We believe that in order to improve clarity for the readers, it is best to focus first on the physiological effects observed in the $\Delta flhE$ mutant, followed by our observations of periplasmic flagellar filament formation in the $\Delta flgHI$ mutant, which explains the cell morphology defect.

318, class III genes?

RE: We modified the sentence as suggested.

330, there are only extracellular filaments shown, maybe mention that the staining procedure would not target periplasmic flagella. Otherwise, I would expect to have an image. Or refer to later.

RE: the text at line 334 was modified as suggested by the reviewer. "If those filaments assemble in the periplasm, it is possible that the staining procedure used here did not permit their observation, notably due to the high molecular weight of antibodies. Extracellular flagellar filament assembly was also observed in either the P-ring (*LflgI*) or L-ring (*LflgH*) *LflhE* mutant background (Fig. S7a)."

362, after the comma, it's not clear which strain is meant.

RE: We modified the sentence: "In absence of inducer and flagella formation, the *LfliC* strain displayed no growth defect (Fig. S7d), whereas complementation with full-length flagellin restored the growth defect in presence of flagella assembly (*LfliC* / *fliC*⁺)."

388, add 'mutants' in front of 'compared'

RE: We modified the sentence as suggested.

426/Fig. 4a, the image is rather small although the result shown is quite important. Maybe it can be enlarged.

RE: We increased the size of Fig. 4a and Fig. 4b.

439, the different helical forms of the Salmonella flagellum are rather complicated matter that come out of nowhere here. Please add two sentences for explanation and a reference.

RE: We improved the text at line 444 to better explain the rationale of investigating effects of different helical form of the flagellar filament on the observed abnormal cell morphology. We further added the following references, which describe the used flagellin point mutants: PMID: **2051483** / PMID: **12904785**

"We then wondered if the change of curvature was caused by the natural helical conformation of the flagellar filament, and if modifying this conformation from helical to straight, curly or coiled would impact the shape of the cells. To do so, we used previously described point mutants of the flagellin FliC, which modify the conformation of the flagellar filaments^{48,49}. Interestingly, none of the point mutants prevented the change in the cell curvature, nor the growth defect observed in *LflgHI LflhE* mutant, suggesting that the helical shape of the

flagellar filament was not responsible for the observed increase of cell curvature itself (Fig. S9)."

484, rather ,spatial control of PG assembly'; see above

RE: We modified the sentence as suggested.

539, as mentioned before, are there any interactions shown? Rod subunits, rod tip (FlgJ), FlhE? Has this been tried?

RE: As mentioned above in our response to reviewer 1, we did not perform biochemical experiments to investigate a putative interaction between the rod subunits and FlhE so far. However, we note that FlhE was previously detected in purified hook-basal bodies by Lee et al (PMID: **22435757**), demonstrating that FlhE is part of the HBB structure. However, the recently published cryo-EM structure of the basal body lacking hooks by Johnson *et al*, (PMID: **33931760**) did not contain any density that could be attributed to FlhE. This suggests that FlhE interaction with the basal body is transient, which is consistent with the model we are proposing in this manuscript. We agree with the reviewer that future studies should focus to biochemically investigate putative interactions between FlhE and the subunits of the rod.